# Neural Adaptive Funnel Dynamic Surface Control with Disturbance-Observer for the PMSM with Time Delays

**DOI:** 10.3390/e24081028

**Published:** 2022-07-26

**Authors:** Menghan Li, Shaobo Li, Junxing Zhang, Fengbin Wu, Tao Zhang

**Affiliations:** 1School of Mechanical Engineering, Guizhou University, Guiyang 550025, China; limenghan1226@163.com (M.L.); zhangtao_202102@163.com (T.Z.); 2State Key Laboratory of Public Big Data, Guizhou University, Guiyang 550025, China; jx_zhanggz@163.com; 3School of Computer Science and Technology, Guizhou University, Guiyang 550025, China; wfaceboss@163.com

**Keywords:** disturbance observer, dynamic surface control, permanent magnetic synchronous motor, funnel control, radial basis function neural networks

## Abstract

This paper suggests an adaptive funnel dynamic surface control method with a disturbance observer for the permanent magnet synchronous motor with time delays. An improved prescribed performance function is integrated with a modified funnel variable at the beginning of the controller design to coordinate the permanent magnet synchronous motor with the output constrained into an unconstrained one, which has a faster convergence rate than ordinary barrier Lyapunov functions. Then, the specific controller is devised by the dynamic surface control technique with first-order filters to the unconstrained system. Therein, a disturbance-observer and the radial basis function neural networks are introduced to estimate unmatched disturbances and multiple unknown nonlinearities, respectively. Several Lyapunov-Krasovskii functionals are constructed to make up for time delays, enhancing control performance. The first-order filters are implemented to overcome the “complexity explosion” caused by general backstepping methods. Additionally, the boundedness and binding ranges of all the signals are ensured through the detailed stability analysis. Ultimately, simulation results and comparison experiments confirm the superiority of the controller designed in this paper.

## 1. Introduction

The permanent magnet synchronous motor (PMSM) is a separately excited generator composed of a stator and a rotator. In recent decades, PMSMs are widely used in aerospace, defense, and other major fields due to the advantages of simple structure, small size, and low noise [1,2]. With the purpose of environmental protection, PMSMs are currently also used in the fields of new energy vehicles, and other major fields [3,4]. However, PMSMs are systems with nonlinearities, strong coupling, and time-varying. Hence, there has been considerable interest in the high-precision control of PMSMs, which is of great help to enhance the aviation industry and ecological conservation for a nation.

The most classical method of controlling PMSMs is backstepping control. The backstepping method splits the n-order complicated system into n subsystems and realizes the virtual control of every subsystem. In the former n−1 steps, it will derive a virtual controller in every step, significantly simplifying the computational process of the controller. Nevertheless, the backstepping method also suffers from the “explosion of complexity” caused by the repeated derivatives of the virtual controller, owing to its inherent properties of it. To circumvent this obstacle, a first-order filter combined with the backstepping control approach [5] is created by Swaroop, called dynamic surface control [6]. Despite dynamic surface control methods [7,8,9], that can reduce computing efforts, various nonlinear factors such as time delays, external disturbances, and physical constraints are ubiquitous in real industrial scenarios [10,11], which may diminish the controlling precision of the PMSM systems. Researchers have proposed proportional integral derivative (PID) control [12], neural network (NN) [5], time delay control [13,14], disturbance observer (DO) [15,16], and constraint control [17,18] methods for different nonlinearities to reach satisfying control results. Hence, the key point is how to design an effective controller to address the various nonlinear uncertainties such as unknown functions, mismatched disturbance, state constraints, and time delays.

For the unknown characteristics and uncertain disturbances, the NN-based adaptive control that has superb evaluating abilities are employed to obtain the high-performance control of nonlinear systems [19,20]. For instance, the authors in [19] design a radial basis function NN (RBFNN) to estimate the error of the high-gain observer and the lumped interference. The authors in [20] utilize a feedforward artificial NN to renew the training parameters of the proportion integration differential controller, offering high dynamic performance. Due to their advantages, RBFNNs are introduced in this paper to estimate the unknown uncertainties. To achieve advanced transient and steady-state performance, NNs still need to be combined with other control strategies.

For the uncertain load disturbances, considerable pioneering investigations based on the DO have been frequently suggested to achieve the steady operation of the PMSM [21]. Based on the sliding mode control technology, a DO is proposed to attain desired anti-load-disturbance capability [11]. A DO is fused into the super-twisting sliding mode method to compensate for the lumped disturbance [22]. A second-order DO is utilized to estimate the parameters perturbations, ensuring the accuracy of the PMSM system [16]. Though the DO-based schemes can effectively suppress the uncertain perturbations, the transient performance that is significant for the robustness of the system is not negligible. To reduce the calculation burdens and further enhance the transient performance, this paper introduces a finite-time second-order command filter to approximate the matched and mismatched disturbance in a finite time with the aid of these investigations.

The physical constraints complicate controller design process, proper strategies which are proposed by researchers to improve the transient performance and stability of nonlinear systems. They have introduced barrier Lyapunov functions [23], prescribed performance functions (PPF) [24,25], and funnel control strategies. As the novel control methods developed, the barrier Lyapunov functions need to be modified to suit specific changing situations, resulting in its non-universality. Likewise, the demand for precise initial values limits the applications of PPF strategies. Based on these, the improved PPFs are applied to further attain superior transient performance [26,27]. Compared with PPFs and barrier Lyapunov functions, the funnel control method is considered a promising way to deal with the constraints owing to its effectiveness for the output overshoot [28,29], and does not need precise initial conditions. The funnel control method has been applied in vehicles [30], and robots [31] because it can avoid the modification of the controllers. According to the above investigations, this paper suggests a funnel controller for the output-constrained PMSM system to achieve excellent control performance.

It is known that time delays harm the dynamic performance and the stability of the PMSMs. The time delay control strategies that can strengthen both transient and steady-state responses, recently, have been involved in many pioneering backstepping investigations. The studies can be roughly divided into two groups namely Smith predictor control tools [32,33] and Lyapunov-Krasovskii functional methods [34,35]. For example, a smith predictor is combined with the speed controller to eliminate the time delays [32]. An appropriate Lyapunov-Krasovskii functional is employed to ensure the asymptotic stability of the delay-dependent PMSM system [34]. A hyperbolic function is utilized to deal with the delay caused by the low-pass filter [36]. From these, a proper Lyapunov-Krasovskii functional is designed in this paper to address the time delay issues, which enhances both the transient and steady-state performances of PMSM.

Motivated by these discussions, a neural adaptive funnel dynamic surface control (FDSC) scheme combined with the DO is designed in this article to address the output-restrained system with time delays and load interference. The primary contributions of this paper are outlined as:(a)The prescribed performance control method can ensure the tracking error converges to a predefined arbitrary small residual set [24,25]. However, the transient performance still needs to be improved. Upon this, a neural adaptive funnel control strategy with advanced transient performance is proposed to restrict the output response of PMSM into a certain funnel region. The simulation section compares the funnel control method with neural dynamic surface control (NDSC) and PID methods, and the funnel controller has smaller tracking error and better transient performance. The advanced transient performance makes the devised controller more applicable.(b)The constraint-considered controller can reasonably simulate the physical constraints in the actual operating environment of the PMSM [23,25]. Nevertheless, there are many other nonlinear factors in the actual industries that need to be taken into account. The FDSC method considers output constraints, time delays, and mismatched external load interference, which allows FDSC to simulate the actual situation more realistically. The Lyapunov-Krasovskii functionals and DO are devised to suppress the time delays and approximate the unmatched external load interference. The simulation section shows that the FDSC method has smaller steady-state and transient errors, and the robustness and dynamic performances are strengthened compared with the NDSC and the PID schemes.

Notation: For any variable c,$\hat{c}$ denotes the estimated value of c, $\tilde{c}=c-\hat{c}$ represents the state error. ‖●‖ indicates the 2-norm of ●.

## 2. System Formulation and Preliminaries

### 2.1. System Statement

Under the (d−q) coordinate frame, formulate the PMSM mathematical model as [7]: (1)$$\left\{ \begin{array}{l} \dot{\theta }=\omega \\ \dot{\omega }=3n_{p}/\left(2J\right)\left[(L_{d}-L_{q})i_{d}i_{q}+\varphi i_{q}\right]-B\omega /J-T_{L}/J\\ {\dot{i}}_{q}=-R_{s}i_{q}/L_{q}-n_{p}L_{d}i_{d}\omega /L_{q}-n_{p}\varphi \omega /L_{q}+u_{q}/L_{q}\\ {\dot{i}}_{d}=-R_{s}i_{d}/L_{d}+n_{p}L_{q}i_{q}\omega /L_{d}+u_{d}/L_{d}\;\;, \end{array}\right.$$
where $u_{q}$ and $u_{d}$ are control variables. The descriptions of PMSM parameters are exhibited in Table 1.

To simplify (1), define $a_{1}=3n_{p}\varphi /2,\;a_{2}=3n_{p}(L_{d}-L_{q})/2,\;b_{1}=\;-R_{s}/L_{q},\;b_{2}=-n_{p}L_{d}/L_{q},\;b_{3}=-n_{p}\varphi /L_{q},\;b_{4}=1/L_{q},\;c_{1}=-R_{s}/L_{d},\;c_{2}=n_{p}L_{q}/L_{d},\;c_{3}=1/L_{d}.$

Moreover, describe the states variables as: (2)$$x_{1}=\theta ,\;x_{2}=\omega ,\;x_{3}=i_{q},\;x_{4}=i_{d}.$$

Considering time delays and asymmetric output constraints, the model (1) can be reconstructed as: (3)$$\{\begin{array}{l} {\dot{x}}_{1}=x_{2}+\Delta l_{1}\left[x(t-\tau_{1})\right]\\ {\dot{x}}_{2}=\frac{a_{1}}{J}x_{3}+\frac{a_{2}}{J}x_{3}x_{4}-\frac{B}{J}x_{2}-\frac{T_{L}}{J}+\Delta l_{2}\left[x(t-\tau_{2})\right]+\Delta E\\ {\dot{x}}_{3}=b_{1}x_{3}+b_{2}x_{2}x_{4}+b_{3}x_{2}+b_{4}u_{q}+\Delta l_{3}\left[x(t-\tau_{3})\right]\\ {\dot{x}}_{4}=c_{1}x_{4}+c_{2}x_{2}x_{3}+c_{3}u_{d}+\Delta l_{4}\left[x(t-\tau_{4})\right], \end{array}$$
where $x={\left(x_{1},x_{2},x_{3},x_{4}\right)}^{T}\in R^{4}$, and the state variable $x_{1}$ subjects to: (4)$$x_{1}\in \Pi_{x_{1}}:=\{x_{1}\in R:\;\;\;x_{d}(t)-f_{1}(t)\;<x_{1}(t)<x_{d}(t)+f_{1}(t)\}.$$

**Remark** **1.***In the actual PMSM operating environment, there are plenty of unknown uncertainties, such as time delays, physical constraints, and matched and mismatched disturbances. It is essential to consider these limitations in the PMSM system to guarantee that the controller designed in this paper approach reality. It is worth mentioning that for the PMSM systems, the unmatched interference*ΔE*and the time delays* $\Delta l_{i}(x(t-\tau_{i})),i=1,\cdots ,4$*in (3) as well as the time-varying symmetric output constraints in (4) are firstly considered simultaneously. Thereby, the controller devised here based on (3) is more suitable for real industrial fields.*

Based on the above observations, the control objective of this paper is to devise a neural adaptive funnel dynamic surface controller based on DO for (3) to realize: (a)The system output $x_{1}$ tracks the desired signal $x_{d}$, where the transient control behavior of (3) can be retained by (4).(b)The other signals in the resulting system are bounded.

To ensure these objectives, the following assumption and lemmas are provided.

**Assumption** **1**([37])**.** *The continuous desired signal* $x_{d}(t)$
*and its ith-order derivatives* $x_{d}^{\left(i\right)}(t),\;\;\;(i=0,\ldots ,4)$
*are bounded. The continuous state-constrained functions* $f_{1}(t)$
*as well as its jth-order derivatives* $f_{1}^{\left(j\right)}(t),\;\;\;(j=0,\ldots ,4)$
*are bounded.*

**Lemma** **1**([38])**.** *For any* $f(\eta_{1},\ldots ,\eta_{n}):R^{m_{r}}\times \ldots \times R^{m_{n}}\to R,$
*there are smooth functions* $\omega_{i}(\eta_{i})>0:\;$
$R^{m_{i}}\to R$
*satisfying* $|f(\eta_{1},\ldots ,\eta_{n})|\leq \sum_{i=1}^{n}\omega_{i}(\eta_{i}).$
*For the initial value* f(0,…,0)=0, $\omega_{i}(0)=0$
*fulfills* $\left|f(0,\ldots ,0)\right|\leq \sum_{i=1}^{n}\omega_{i}(0)$
*as well*.

**Remark** **2.***According to Lemma 1, there are continuous positive functions* $\xi_{ik},i=1,\ldots ,4$*restricting the time-delay terms* $\Delta l_{i}(x(t-\tau_{i})),i=1,\ldots ,4$*in the system (3) as* $\Delta l_{i}(x(t-\tau_{i}))$$\leq \sum_{k=1}^{4}\xi_{ik}(x_{k}(t-\tau_{i}))$*. With the aid of Young’s inequality, it leads to: *


(5)
$$e_{i}\Delta l_{i}\left[x(t-\tau_{i})\right]\leq 1/2ne_{i}^{2}+1/2\sum_{j=1}^{n}\xi_{ij}^{2}(x_{j}(t-\tau_{i})),i=1,\ldots ,4.\;$$


**Lemma** **2**([39,40])**.** *For any real variables* p,q*, and the positive constants* $m_{i},\;\;i=1,2,3$*, the following inequality holds: *


(6)
$${\left|a\right|}^{m_{1}}{\left|b\right|}^{m_{2}}⩽\frac{m_{1}}{m_{1}+m_{2}}m_{3}{\left|a\right|}^{m_{1}+m_{2}}+\frac{m_{2}}{m_{1}+m_{2}}m_{3}^{-m_{1}/m_{2}}{\left|b\right|}^{m_{1}+m_{2}}.\;$$


**Lemma** **3**([41])**.** *For* σ>0,
*there exists the set* $\Omega_{e}:=\{e\in R:\left|e\right|\leq 0.2554\sigma \}.$
*For* $e\notin \Omega_{e}$*, the inequality* $1-16{tanh}^{2}(e/\sigma )<0$
*holds.*

Thereafter, function arguments are sometimes dropped without confusion.

### 2.2. Neural Network Systems and Function Approximation

Since the RBFNNs can approximate the unknown hard-to-calculated functions that are in a closed set at any precision [42,43], this paper introduces an RBFNN function to estimate unknown functions W(X): (7)$$W(X)={℘}^{*T}P(X)+\psi (X),\;\;\forall X\in \Omega_{X},$$
where $X={\left[x_{1},x_{2},\cdots ,x_{n}\right]}^{T}$ indicates the input vector, ψ(X) fulfills $\left|\psi (X)\right|<\psi_{M}$ with a bounded constant $\psi_{M}$. P(X)=${\left[p_{1}(x),p_{2}(x),\ldots ,p_{l}(x)\right]}^{T}$ is a vector of basis function, and select $p_{i}(x)$ as the versatile Gaussian functions: (8)$$p_{i}(x)=\exp\left[-{\left(x-\nu_{i}\right)}^{T}(x-\nu_{i})/\chi_{i}^{2}\right],\;\;i=1,2,\ldots ,m.$$

Construct the expected weight vector ${℘}^{*}$ as: (9)$${℘}^{*}=\arg\;\;\underset{℘\in \;\;R^{n}}{\min}\left\{\underset{X\in \;\;D_{X}}{\sup}\left\|W(X)-{\hat{℘}}^{T}P(X)\right\|\right\}.$$

The 2-norms of the variables are utilized to assess the weights allowing for a reduction in the computational burden of RBFNN [42,43]. Consequently, it leads to: (10)$$\beta_{i}={\left\|{℘}_{i}\right\|}^{2}={℘}_{i}^{T}{℘}_{i}\;,\;\;\;\;\;i=1,\ldots ,4.$$

## 3. Design of Neural Adaptive Funnel Control

This section introduces the funnel controller designed in this paper. In Section 3.1, a funnel-type variable with improved PPF is introduced to remove the output constraint of the variable $x_{1}$. In Section 3.2, a DO is proposed to approximate the matched and mismatched disturbance that is utilized in step 2. In Section 3.3, four steps are listed to show the whole design procedure of the funnel controller for the PMSM system.

### 3.1. Funnel Control with Improved Prescribed Performance Function

The core point of the funnel control is devising the given functions of the envelope as the boundaries that restrict the tracking error $s_{1}(t)$, where $s_{1}(t)=x_{1}-x_{d}$. Based on this idea, a funnel-type function is selected at the very beginning of the controller design as: (11)$$g(t)=F_{k}\left(f_{1}(t),G(t),\left\|s_{1}(t)\right\|\right).$$

A PPF $f^{*}(t)$ [44] is constructed to obtain both steady-state and transient performances of $s_{1}(t)$: (12)$$f^{*}(t)=(f_{0}-f_{\infty })\exp(-\pi t)+f_{\infty },$$
where the design parameters 3 $f_{0}>f_{\infty }>0$ and the minimum convergence rate value π>0.

From [44], we can attain that the control objectives can be achieved when the following condition satisfied: (13)$$\left|s_{1}\left(t\right)\right|<\omega f^{*}(t),$$
where the design constants ω>0.

To obtain better steady performance, a funnel control with envelope boundaries is introduced. An improved PPF based on (12) is proposed: (14)$$f_{1}(t)=f_{0}\exp(-\pi t)+\frac{t}{\pi (t+1)}f_{\infty },$$
where $f_{\infty }/\pi$ represents the steady-state error.

**Remark** **3.***The improved PPF* $f_{1}(t)$*indicated by (14) has the same properties as (12). Moreover, the improved PPF has a faster convergence speed and more outstanding transient performance than the conventional PPF while choosing the same parameters values of*$f_{0},f_{\infty },\pi$*. As an example, Figure 1 gives the profiles of (14) and (12) with* $f_{0}=1,f_{\infty }=0.05,\pi =3,5,10$.

The gain function $F_{k}(\cdot )$ [31] can be regulated by: (15)$$F_{k}\left(f_{1}(t),G(t),\left\|s_{1}(t)\right\|\right)=G(t)/\left(f_{1}(t)-\left\|s_{1}(t)\right\|\right).$$

It can be concluded from (15) that the value of $F_{k}(\cdot )$ decreases as $f_{1}(t)$ moves away from $\left\|s_{1}(t)\right\|$ when G(t) is supposed to be fixed. Regrettably, the gain $F_{k}(\cdot )$ formulated by (15) is not differentiable at the point of $s_{1}(t)=0$, which does not meet the using conditions of backstepping.

To circumvent the obstacle, a modified funnel variable $\eta_{1}$ by virtue of (14) and [31] is first created as follows: (16)$$\eta_{1}=s_{1}^{2}/\left(f_{1}^{2}-s_{1}^{2}\right).$$

**Remark** **4.***If* $f_{1}\to s_{1}$*, the value of* $\eta_{1}$*will be infinite. For this reason, assuming that the starting value of the tracking error* $s_{1}(0)$*is limited by the funnel boundaries. Choosing a proper design parameter* $f_{0}$*can effectively eliminate the infinite starting value.*

**Remark** **5.**
*Compared with prescribed performance control [45], the improved PPF-based funnel control not only provides better transient performance but is also simpler and more efficient without calculating inverse conversion errors, which is conducive to the stable operation of PMSM.*


The time derivatives of $\eta_{i}$ in (16) are obtained as: (17)$${\dot{\eta }}_{1}=\frac{2s_{1}f_{1}^{2}}{{\left(f_{1}^{2}-s_{1}^{2}\right)}^{2}}\left({\dot{s}}_{1}-s_{1}\frac{{\dot{f}}_{1}}{f_{1}}\right)=\Gamma_{1}\left({\dot{s}}_{1}-s_{1}\frac{{\dot{f}}_{1}}{f_{1}}\right),$$
where the variable $\Gamma_{1}=2s_{1}f_{1}^{2}/{\left(f_{1}^{2}-s_{1}^{2}\right)}^{2}$.

### 3.2. Disturbance Observer

For the difficult-to-compute external disturbance, a DO is an effective utensil to observe the matched and mismatched interference. Hence, introduce the intermediate variables d,D, and conceive the DO as [46]: (18)$$\left\{ \begin{array}{l} {\dot{\hat{x}}}_{2}=a_{1}/Jx_{3}+a_{2}/Jx_{3}x_{4}-B/Jx_{2}-T_{L}/J+\Delta l_{2}(x(t-\tau_{2}))+\Delta E_{1}\\ \Delta E_{1}=-\kappa_{1}\iota_{1}^{1/3}{\left|{\hat{x}}_{2}-x_{2}\right|}^{2/3}\mathrm{sgn}({\hat{x}}_{2}-x_{2})+\Delta {\hat{E}}_{2}\\ \Delta {\dot{\hat{E}}}_{2}=-\kappa_{1}\iota_{1}^{1/2}{\left|\Delta {\hat{E}}_{2}-\Delta E_{2}\right|}^{1/2}{\mathrm{sgn}}^{1/2}(\Delta {\hat{E}}_{2}-\Delta E_{2})+\Delta \hat{E}\\ \Delta \dot{\hat{E}}=-\kappa_{2}\iota_{1}\mathrm{sgn}(\Delta \hat{E}-\Delta {\dot{\hat{E}}}_{2}), \end{array}\right.$$
where coefficients $\kappa_{1}>0,\kappa_{2}>0,\iota_{1}>0$.

Describing variables as ${\tilde{x}}_{2}={\hat{x}}_{2}-x_{2},\Delta \tilde{E}=\Delta \hat{E}-\Delta E,\Delta {\tilde{E}}_{2}=\Delta {\hat{E}}_{2}-\Delta E_{2}$, it leads to: (19)$$\left\{ \begin{array}{l} {\dot{\tilde{x}}}_{2}=-\kappa_{1}\iota_{1}^{1/3}{\left|{\hat{x}}_{2}-x_{2}\right|}^{2/3}\mathrm{sgn}({\hat{x}}_{2}-x_{2})+\Delta \tilde{E}\\ \Delta \dot{\tilde{E}}=-\kappa_{1}\iota_{1}^{1/2}{\left|\Delta {\hat{E}}_{2}-\Delta E_{2}\right|}^{1/2}\mathrm{sgn}(\Delta {\hat{E}}_{2}-\Delta E_{2})+(\Delta \hat{E}-\Delta E)\\ \Delta \tilde{E}\in -\kappa_{2}\iota_{1}\mathrm{sgn}(\Delta \hat{E}-\Delta {\dot{\hat{E}}}_{2})+[-\iota_{1},\iota_{1}]. \end{array}\right.$$

Supposing a negligible rate of change for load disturbance $\Delta E_{2}$, then it is bounded. According to $\Delta {\tilde{E}}_{2}=\Delta {\hat{E}}_{2}-\Delta E_{2}$, it can be concluded that $\Delta \tilde{E}$ is bounded. Similarly, we can attain that $\Delta \hat{E}$ and ${\hat{x}}_{2}$ are limited to the small adjacency of ΔE and $x_{2}$, respectively.

**Remark** **6.***The interference error can be well estimated by the excellent DO (18). In other words, we can select suitable parameters*$\kappa_{1},\kappa_{2}$*and* $\iota_{1}$*to tackle the matched and unmatched load disturbance. Moreover, the excellent DO can prevent the system tremors due to the symbolic function in the classical sliding mode control method simultaneously [21].*

### 3.3. Neural Adaptive Funnel Controller Design

This subsection will provide a dynamic surface controller by integrating the funnel control technology. First of all, let us introduce the following coordinate conversion: (20)$$e_{1}=\eta_{1},\;e_{2}=x_{2}-u_{2c},\;e_{3}=x_{3}-u_{3c},\;e_{4}=x_{4}.$$

Define the first-order filters as: (21)$$\lambda_{i}{\dot{u}}_{ic}+u_{ic}=u_{i},\;u_{ic}(0)=u_{i}(0),\;i=2,3,$$
where $\lambda_{i}$ are constants.

**Remark** **7.***This paper introduces first-order filters to bypass the repeated derivatives of* $u_{i}$*so that the “explosion of complexity” can be eliminated by designing proper filters. However, there exist filter errors that influence the control precision by introducing the first-order filters into the controller design procedure. When choosing proper Lyapunov functions to devise the virtual and actual controller, the filter errors need to be taken into account to improve the control accuracy.*

Akin to (20), describe the filter errors $\phi_{i}$ as: (22)$$\phi_{i}=u_{ic}-u_{i}\;,\;i=2,3.$$

Fusing (3) and (16) into (20), and take the derivatives of $e_{i},i=1,\ldots ,4$ in (20): (23)$$\left\{ \begin{array}{l} {\dot{e}}_{1}=\Gamma_{1}\left\{e_{2}+\phi_{2}+u_{2}-{\dot{x}}_{d}+\Delta l_{1}\left[x(t-\tau_{1})\right]-s_{1}{\dot{f}}_{1}/f_{1}\right\}\\ {\dot{e}}_{2}=e_{3}+\phi_{3}+u_{3}+\left(\frac{a_{1}}{J}-1\right)x_{3}+\frac{a_{2}}{J}x_{3}x_{4}-\frac{B}{J}x_{2}-\frac{T_{L}}{J}-{\dot{u}}_{2c}+\Delta l_{2}\left[x(t-\tau_{2})\right]+\Delta E\\ {\dot{e}}_{3}=b_{1}x_{3}+b_{2}x_{2}x_{4}+b_{3}x_{2}+b_{4}u_{q}\;\;+\Delta l_{3}\left[x(t-\tau_{3})\right]-{\dot{u}}_{3c}\\ {\dot{e}}_{4}=c_{1}x_{4}+c_{2}x_{2}x_{3}+c_{3}u_{d}+\Delta l_{4}\left[x(t-\tau_{4})\right].\; \end{array}\right.$$

This paper utilized RBFNNs to estimate the unknown functions, so there exist estimation errors that need to be computed. Describe the approximated errors ${\tilde{\beta }}_{i}$ as: (24)$${\tilde{\beta }}_{i}=\beta_{i}-{\hat{\beta }}_{i},i=1,\ldots ,4.$$

The next steps are further design procedures of the neural adaptive FDSC. The brief structure of the control procedure is shown in Figure 2.
**Step 1.** The design of the stunning control law $u_{2}$ and the adaptive law ${\dot{\hat{\beta }}}_{1}$.

Taking the tracking error $e_{1}$, filter error $\phi_{2}$, approximation error ${\tilde{\beta }}_{1}$, and the Lyapunov–Krasovskii functional $V_{T}$ into account, choose the Lyapunov function $V_{1}$ as: (25)$$V_{1}=1/2e_{1}^{2}+1/2\phi_{2}^{2}+1/\left(2d_{1}\right){\tilde{\beta }}_{1}^{2}+V_{T},$$
subject to the Lyapunov–Krasovskii functional $V_{T}$ as: (26)$$V_{T}=\frac{1}{2}\sum_{i=1}^{4}\sum_{j=1}^{4}\exp\left[-ƛ\left(t-\tau_{i}\right)\right]\int_{t-\tau_{i}}^{t}\exp\left(ƛv\right)\xi_{ij}^{2}\left[x_{j}(v)\right]dv,$$
where the design constants $d_{1}>0,ƛ>0$, $\xi_{ij}$ are utilized to address the time delays below.

**Remark** **8.***To further improve the precision of the controller designed later, the**proper Lyapunov-Krasovskii functional* $\;V_{T}$*is utilized to compensate for the time-varying delays.*

Taking the derivative of $V_{T}$ in (26) obtains: (27)$${\dot{V}}_{T}=\frac{1}{2}\sum_{i=1}^{4}\sum_{j=1}^{4}\exp\left[-ƛ\left(t-\tau_{i}\right)\right]\xi_{ij}^{2}\left[x_{j}(t)\right]-\frac{1}{2}\sum_{i=1}^{4}\sum_{j=1}^{4}\xi_{ij}^{2}\left[x_{j}(t-\tau_{i})\right]-ƛV_{T}.$$

With (24) and (27), deriving $V_{1}$ in (25) yields: (28)$${\dot{V}}_{1}=e_{1}{\dot{e}}_{1}-\frac{1}{d_{1}}{\tilde{\beta }}_{1}{\dot{\hat{\beta }}}_{1}+\frac{1}{2}\sum_{i=1}^{4}\sum_{j=1}^{4}\exp\left[-ƛ\left(t-\tau_{i}\right)\right]\xi_{ij}^{2}\left[x_{j}(t)\right]-\frac{1}{2}\sum_{i=1}^{4}\sum_{j=1}^{4}\xi_{ij}^{2}\left[x_{j}(t-\tau_{i})\right]+\phi_{2}{\dot{\phi }}_{2}-ƛV_{T}.$$

Fusing (23) into (28) generates: (29)$$\begin{array}{c} \;{\dot{V}}_{1}=e_{1}\Gamma_{1}\left\{e_{2}+\phi_{2}+u_{2}-{\dot{x}}_{d}+\Delta l_{1}\left[x(t-\tau_{1})\right]-s_{1}\frac{{\dot{f}}_{1}}{f_{1}}\right\}-\frac{1}{2}\sum_{i=1}^{4}\sum_{j=1}^{4}\xi_{ij}^{2}\left[x_{j}(t-\tau_{i})\right]\\ +\frac{1}{2}\sum_{i=1}^{4}\sum_{j=1}^{4}\exp\left[-ƛ\left(t-\tau_{i}\right)\right]\xi_{ij}^{2}\left[x_{j}(t)\right]-\frac{1}{d_{1}}{\tilde{\beta }}_{1}{\dot{\hat{\beta }}}_{1}-ƛV_{T}+\phi_{2}{\dot{\phi }}_{2}. \end{array}$$

Based on Remark 2 with n=4, the following inequality can be obtained: (30)$$e_{1}\Gamma_{1}\Delta l_{1}\left[x(t-\tau_{1})\right]\leq 2e_{1}^{2}\Gamma_{1}^{2}+\frac{1}{2}\sum_{j=1}^{4}\xi_{1j}^{2}\left[x_{j}(t-\tau_{1})\right].$$

Taking (30) into (29) derives: (31)$${\dot{V}}_{1}\leq e_{1}\Gamma_{1}\left(e_{2}+\phi_{2}+u_{2}-{\dot{x}}_{d}-s_{1}\frac{{\dot{f}}_{1}}{f_{1}}\right)-\frac{1}{2}\sum_{i=2}^{4}\sum_{j=1}^{4}\xi_{ij}^{2}\left[x_{j}(t-\tau_{i})\right]-\frac{1}{d_{1}}{\tilde{\beta }}_{1}{\dot{\hat{\beta }}}_{1}+2e_{1}^{2}\Gamma_{1}^{2}+T\left(x\right)+\phi_{2}{\dot{\phi }}_{2}-ƛV_{T},$$
where the time delay function $T(x)=1/2\sum_{i=1}^{4}\sum_{j=1}^{4}\left(-\exp(ƛ\tau_{i})\xi_{ij}^{2}(x_{j}(t))\right)$. We can infer that $\lim_{e_{1}\to 0}(T(x)/e_{1})\to \infty$. Consequently, $(16/e_{1})\tanh^{2}(e_{1}/\sigma_{1})T$ is brought in (32) to further eliminate the time delays. Then, rewrite (31) as: (32)$$\begin{array}{c} {\dot{V}}_{1}\leq e_{1}\left[\Gamma_{1}e_{2}+\Gamma_{1}\phi_{2}+\Gamma_{1}u_{2}+\frac{16}{e_{1}}T\tanh^{2}\left(\frac{e_{1}}{\sigma_{1}}\right)+2e_{1}\Gamma_{1}^{2}-\Gamma_{1}{\dot{x}}_{d}-\Gamma_{1}s_{1}\frac{{\dot{f}}_{1}}{f_{1}}\right]\\ +\phi_{2}{\dot{\phi }}_{2}-ƛV_{T}+\left[1-16\tanh^{2}\left(\frac{e_{1}}{\sigma_{1}}\right)\right]T-\frac{1}{2}\sum_{i=2}^{4}\sum_{j=1}^{4}\xi_{ij}^{2}\left[x_{j}(t-\tau_{i})\right]-\frac{1}{d_{1}}{\tilde{\beta }}_{1}{\dot{\hat{\beta }}}_{1}.\; \end{array}$$

Then this paper utilizes an unknown function $W_{1}(X_{1})$ to generalize the unknown functions: (33)$$W_{1}(X_{1})=3e_{1}\Gamma_{1}^{2}+e_{1}+16/e_{1}T\tanh^{2}\left(e_{1}/\sigma_{1}\right)-\Gamma_{1}{\dot{x}}_{d},$$
where $W_{1}(X_{1})$ is an unknown function. According to (11), (16) and (20), it can be learned that $e_{1}$ is composed of a known desired signal $x_{d}$ and an unknown state $x_{1}$, $\Gamma_{1}$ consists of a known desired signal $x_{d}$, an assured function $f_{1}$, and an unknown state $x_{1}$. The RBFNN is developed to estimate the unknown function $W_{1}(X_{1})$, then $X_{1}={\left[x_{1},\ldots ,x_{4},x_{d},{\dot{x}}_{d}\right]}^{T}$.

For the unknown and uncertain functions, RBFNN is considered as a promising tool to approximate them at any precision. An RBFNN is designed to approximate $W_{1}(X_{1})$: (34)$$W_{1}(X_{1})={℘}_{1}^{T}P_{1}(X_{1})+\psi_{1}(X_{1}),\;\;\;\left|\psi_{1}(X_{1})\right|⩽\psi_{M},$$
where $\psi_{M}$ stands for the positive bounded constant.

As it can be seen from (33), (32) can be simplified by an RBFNN: (35)$$\begin{array}{c} {\dot{V}}_{1}\;\leq e_{1}\left(\Gamma_{1}e_{2}+\Gamma_{1}\phi_{2}+\Gamma_{1}u_{2}+{℘}_{1}^{T}P_{1}(X_{1})+\psi_{1}(X_{1})-\Gamma_{1}s_{1}\frac{{\dot{f}}_{1}}{f_{1}}-e_{1}\Gamma_{1}^{2}-e_{1}\right)-\frac{1}{d_{1}}{\tilde{\beta }}_{1}{\dot{\hat{\beta }}}_{1}\\ +\left[1-16/e_{1}T\tanh^{2}\left(e_{1}/\sigma_{1}\right)\right]T+\phi_{2}{\dot{\phi }}_{2}-ƛV_{T}-\frac{1}{2}\sum_{i=2}^{4}\sum_{j=1}^{4}\xi_{ij}^{2}\left[x_{j}(t-\tau_{i})\right]. \end{array}$$

According to (34) and Lemma 2 with $m_{1}=m_{2}=m_{3}=1$, functions that are difficult to compute can be deflated to: (36)$$\left\{ \begin{array}{l} e_{1}\Gamma_{1}\phi_{2}\leq e_{1}^{2}\Gamma_{1}^{2}+\frac{1}{4}\phi_{2}^{2}\\ e_{1}W_{1}(X_{1})=e_{1}[{℘}_{1}^{T}P_{1}(X_{1})+\psi_{1}(X_{1})]\leq \frac{1}{4\mu_{1}^{2}}\beta_{1}e_{1}^{2}P_{1}^{T}P_{1}+\mu_{1}^{2}+\frac{\psi_{M}^{2}}{4}+e_{1}^{2}, \end{array}\right.$$
where $\mu_{1}$ stands for the positive design constant.

Integrating (36) into (35) generates: (37)$$\begin{array}{ll} {\dot{V}}_{1}\;\leq & e_{1}\left(\Gamma_{1}e_{2}+\Gamma_{1}u_{2}-\Gamma_{1}s_{1}\frac{{\dot{f}}_{1}}{f_{1}}\right)+\left[1-16\tanh^{2}\left(\frac{e_{1}}{\sigma_{1}}\right)\right]T-\frac{1}{2}\sum_{i=2}^{4}\sum_{j=1}^{4}\xi_{ij}^{2}\left[x_{j}(t-\tau_{i})\right]-\frac{1}{d_{1}}{\tilde{\beta }}_{1}{\dot{\hat{\beta }}}_{1}\\ & -ƛV_{T}+\phi_{2}{\dot{\phi }}_{2}+\frac{1}{4\mu_{1}^{2}}e_{1}^{2}\beta_{1}P_{1}^{T}P_{1}+\frac{1}{4}\phi_{2}^{2}+\frac{\psi_{M}^{2}}{4}+\mu_{1}^{2}. \end{array}$$

Devise the stunning control law $u_{2}$ and the adaptive law ${\dot{\hat{\beta }}}_{1}$ as: (38)$$\begin{array}{l} u_{2}=-\frac{(f_{1}^{2}-s_{1}^{2})s_{1}}{2f_{1}^{2}}\left(k_{1}+\frac{1}{4\mu_{1}^{2}}{\hat{\beta }}_{1}P_{1}^{T}P_{1}\right)+s_{1}\frac{{\dot{f}}_{1}}{f_{1}}\\ {\dot{\hat{\beta }}}_{1}=\frac{d_{1}}{4\mu_{1}^{2}}e_{1}^{2}P_{1}^{T}P_{1}-\gamma_{1}{\hat{\beta }}_{1}, \end{array}$$
where the design constants $k_{1}>0,\gamma_{1}>0$.

Inserting the control law $u_{2}$ and adaptive law ${\dot{\hat{\beta }}}_{1}$ in (38) into (37) derives: (39)$$\begin{array}{ll} {\dot{V}}_{1}\;\leq & -k_{1}e_{1}^{2}+\left[1-16\tanh^{2}\left(\frac{e_{1}}{\sigma_{1}}\right)\right]T-\frac{1}{2}\sum_{i=2}^{4}\sum_{j=1}^{4}\xi_{ij}^{2}\left[x_{j}(t-\tau_{i})\right]+\frac{\gamma_{1}}{d_{1}}{\tilde{\beta }}_{1}{\hat{\beta }}_{1}\\ & -ƛV_{T}+\Gamma_{1}e_{1}e_{2}+\phi_{2}{\dot{\phi }}_{2}+\mu_{1}^{2}+\frac{\psi_{M}^{2}}{4}+\frac{\phi_{2}^{2}}{4}. \end{array}$$

According to (21)–(24), and (38), differentiate the filter error $\phi_{2}$ in (22) as: (40)$${\dot{\phi }}_{2}={\dot{u}}_{2c}-{\dot{u}}_{2}=-\frac{\phi_{2}}{\lambda_{2}}+M_{2}(e_{1},e_{2},\phi_{2},{\hat{\beta }}_{1},x_{d},{\dot{x}}_{d},{\ddot{x}}_{d}),$$
where $M_{2}(e_{1},e_{2},\phi_{2},{\hat{\beta }}_{1},x_{d},{\dot{x}}_{d},{\ddot{x}}_{d})\;$ is a smooth function.

In virtual of [47], we can infer that there exists an upper value ${\bar{M}}_{2}({\bar{M}}_{2}\geq 0)$ for the original conditions to restrict $M_{2}(e_{1},e_{2},\phi_{2},{\hat{\beta }}_{1},x_{d},{\dot{x}}_{d},{\ddot{x}}_{d})\;$ within the prescribed set, it produces: (41)$${\dot{\phi }}_{2}\leq -\frac{\phi_{2}}{\lambda_{2}}+{\bar{M}}_{2},$$
where ${\bar{M}}_{2}>\left|M_{2}(e_{1},e_{2},\phi_{2},{\hat{\beta }}_{1},x_{d},{\dot{x}}_{d},{\ddot{x}}_{d})\right|.$

Based on Lemma 2 with $m_{1}=m_{2}=m_{3}=1$, consider $a=\phi_{2},b={\bar{M}}_{2}$. Then, the hard-to-compute function is converted into: (42)$$\phi_{2}{\dot{\phi }}_{2}=-\phi_{2}^{2}/\lambda_{2}+\phi_{2}{\bar{M}}_{2}\leq -\left(1/\lambda_{2}-1/4\right)\phi_{2}^{2}+{\bar{M}}_{2}^{2}.$$

Substituting (42) into (39) generates: (43)$$\begin{array}{ll} {\dot{V}}_{1}\;\leq & -k_{1}e_{1}^{2}-1/2\sum_{i=2}^{4}\sum_{j=1}^{4}\xi_{ij}^{2}\left[x_{j}(t-\tau_{i})\right]-\left(\frac{1}{\lambda_{2}}-\frac{1}{4}\right)\phi_{2}^{2}+\left[1-16\tanh^{2}\left(\frac{e_{1}}{\sigma_{1}}\right)\right]T+\frac{\gamma_{1}}{d_{1}}{\tilde{\beta }}_{1}{\hat{\beta }}_{1}\\ & -ƛV_{T}+\Gamma_{1}e_{1}e_{2}+{\bar{M}}_{2}^{2}+\psi_{M}^{2}/4+\mu_{1}^{2}. \end{array}$$
**Step 2.** The design of the stunning control law $u_{3}$ and the adaptive law ${\dot{\hat{\beta }}}_{2}$.

Considering the state error $e_{2}$, filter error $\phi_{3}$ and the approximation error ${\tilde{\beta }}_{2}$, the second Lyapunov function $V_{2}$ is chosen as: (44)$$V_{2}=V_{1}+1/2e_{2}^{2}+1/2\phi_{3}^{2}+1/\left(2d_{2}\right){\tilde{\beta }}_{2}^{2},$$
where $d_{2}$ denotes the positive constants.

With (24), taking the derivative of $V_{2}$ in (44) leads to: (45)$${\dot{V}}_{2}={\dot{V}}_{1}+e_{2}{\dot{e}}_{2}+\phi_{3}{\dot{\phi }}_{3}-1/d_{2}{\tilde{\beta }}_{2}{\dot{\hat{\beta }}}_{2}.$$

Integrating (23) and (43) into (45) produces: (46)$$\begin{array}{ll} {\dot{V}}_{2}\leq & -k_{1}e_{1}^{2}-1/2\sum_{i=2}^{4}\sum_{j=1}^{4}\xi_{ij}^{2}\left[x_{j}(t-\tau_{i})\right]-\left(\frac{1}{\lambda_{2}}-\frac{1}{4}\right)\phi_{2}^{2}+\left[1-16\tanh^{2}\left(\frac{e_{1}}{\sigma_{1}}\right)\right]T+\frac{\gamma_{1}}{d_{1}}{\tilde{\beta }}_{1}{\hat{\beta }}_{1}\\ & +e_{2}\{e_{3}\;\;\;+\phi_{3}+u_{3}+\left(a_{1}/J-1\right)x_{3}+a_{2}/Jx_{3}x_{4}\;-B/Jx_{2}-T_{L}/J+\Delta l_{2}\left[x(t-\tau_{2})\right]\\ & -{\dot{u}}_{2c}+\Delta E\}-ƛV_{T}+\phi_{3}{\dot{\phi }}_{3}+\Gamma_{1}e_{1}e_{2}+{\bar{M}}_{2}^{2}+\mu_{1}^{2}+\psi_{M}^{2}/4-1/d_{2}{\tilde{\beta }}_{2}{\dot{\hat{\beta }}}_{2}. \end{array}$$

Resembling (30), define n=4, then it derives: (47)$$e_{2}\Delta l_{2}\left[x(t-\tau_{2})\right]\leq 2e_{2}^{2}+\frac{1}{2}\sum_{j=1}^{4}\xi_{2j}^{2}\left[x_{j}(t-\tau_{2})\right].$$

Assume that $\left|\Delta \hat{E}-\Delta E=\Delta \tilde{E}\right|\leq q_{1}$, and the constant $q_{1}>0$. It is hard to determine the positive and negative of $e_{2}$, an inequality $(\Delta \hat{E}-\Delta E)e_{2}\leq \left|\Delta \hat{E}-\Delta E\right|\left|e_{2}\right|\leq q_{1}\left|e_{2}\right|$ is brought to ensure that the following inequality holds. Pouring (47) into (46) gets: (48)$$\begin{array}{ll} {\dot{V}}_{2}\leq & -k_{1}e_{1}^{2}-\frac{1}{2}\sum_{i=3}^{4}\sum_{j=1}^{4}\xi_{ij}^{2}\left[x_{j}(t-\tau_{i})\right]-\left(\frac{1}{\lambda_{2}}-\frac{1}{4}\right)\phi_{2}^{2}+\left[1-16\tanh^{2}\left(\frac{e_{1}}{\sigma_{1}}\right)\right]T+\frac{\gamma_{1}}{d_{1}}{\tilde{\beta }}_{1}{\hat{\beta }}_{1}\\ & +e_{2}[e_{3}\;\;\;+\phi_{3}+\Delta \hat{E}+\left(a_{1}/J-1\right)x_{3}+u_{3}\;-B/Jx_{2}+a_{2}/Jx_{3}x_{4}-T_{L}/J\;+2e_{2}-{\dot{u}}_{2c}\\ & +\Gamma_{1}e_{1}]-ƛV_{T}+\phi_{3}{\dot{\phi }}_{3}+{\bar{M}}_{2}^{2}+q_{1}\left|e_{2}\right|+\mu_{1}^{2}+\psi_{M}^{2}/4-1/d_{2}{\tilde{\beta }}_{2}{\dot{\hat{\beta }}}_{2}.\; \end{array}$$

Design $F_{2}(X_{2})$ as
(49)$$F_{2}(X_{2})=\left(\frac{a_{1}}{J}-1\right)x_{3}+\frac{a_{2}}{J}x_{3}x_{4}-\frac{B}{J}x_{2}-\frac{T_{L}}{J}+4e_{2}+\Gamma_{1}e_{1},$$
where $e_{1}=x_{1}-x_{d},e_{2}=x_{2}-\alpha_{2c}$, $\Gamma_{1}$ is composed of a known desired signal $x_{d}$, a given function $f_{1}$, and an unknown state $x_{1}$. Based on the structure of the uncertain function $F_{2}(X_{2})$, $X_{2}$ is composed of $x_{1},x_{2},x_{3},x_{4},x_{d}$ and $u_{2c}$. Then it leads to $X_{2}={\left[x_{1},\ldots ,x_{4},x_{d},u_{2c}\right]}^{T}$.

A piecewise function $W_{2}(X_{2})$ is devised as: (50)$$W_{2}(X_{2})=\left\{ \begin{array}{l} F_{2}(X_{2})-q_{1},\;\;\;\;\;\;e_{2}\leq 0\\ F_{2}(X_{2})+q_{1},\;\;\;\;\;\;e_{2}>0, \end{array}\right.$$
where $X_{2}={\left[x_{1},\ldots ,x_{4},x_{d},u_{2c}\right]}^{T}$.

Integrating (50) into (48) generates: (51)$$\begin{array}{ll} {\dot{V}}_{2}\leq & -k_{1}e_{1}^{2}-\frac{1}{2}\sum_{i=2}^{4}\sum_{j=1}^{4}\xi_{ij}^{2}\left[x_{j}(t-\tau_{i})\right]-\left(\frac{1}{\lambda_{2}}-\frac{1}{2}\right)\phi_{2}^{2}-\frac{1}{d_{2}}{\tilde{\beta }}_{2}{\dot{\hat{\beta }}}_{2}+\left[1-16\tanh^{2}\left(\frac{e_{1}}{\sigma_{1}}\right)\right]T+\frac{\gamma_{1}}{d_{1}}{\tilde{\beta }}_{1}{\hat{\beta }}_{1}\\ & +e_{2}\left[e_{3}\;\;\;+\phi_{3}+u_{3}+\Delta \hat{E}+W_{2}(X_{2})-2e_{2}-{\dot{u}}_{2c}\right]-ƛV_{T}+\phi_{3}{\dot{\phi }}_{3}+{\bar{M}}_{2}^{2}+\mu_{1}^{2}+\frac{\psi_{M}^{2}}{4}.\; \end{array}$$

Akin to (34), the unknown function $W_{2}(X_{2})$ can be approximated by an RBFNN: (52)$$W_{2}(X_{2})={℘}_{2}^{T}P_{2}(X_{2})+\psi_{2}(X_{2}),\;\;\;\left|\psi_{2}(X_{2})\right|<\psi_{M}.$$

Re-express (51) as: (53)$$\begin{array}{ll} {\dot{V}}_{2}\leq & -k_{1}e_{1}^{2}-\frac{1}{2}\sum_{i=3}^{4}\sum_{j=1}^{4}\xi_{ij}^{2}\left[x_{j}(t-\tau_{i})\right]-\left(\frac{1}{\lambda_{2}}-\frac{1}{2}\right)\phi_{2}^{2}+\left[1-16\tanh^{2}\left(\frac{e_{1}}{\sigma_{1}}\right)\right]T+\frac{\gamma_{1}}{d_{1}}{\tilde{\beta }}_{1}{\hat{\beta }}_{1}\\ & +e_{2}\left[e_{3}\;\;\;+\phi_{3}+u_{3}\;+\Delta \hat{E}+{℘}_{2}^{T}P_{2}(X_{2})+\psi_{2}(X_{2})-{\dot{u}}_{2c}-2e_{2}\right]-ƛV_{T}+\phi_{3}{\dot{\phi }}_{3}+{\bar{M}}_{2}^{2}\\ & +\mu_{1}^{2}+\psi_{M}^{2}/4-1/d_{2}{\tilde{\beta }}_{2}{\dot{\hat{\beta }}}_{2}.\; \end{array}$$

Similar to (36), the inequalities below can be obtained with $m_{1}=m_{2}=m_{3}=1$: (54)$$\left\{ \begin{array}{c} \;\;\;\;\;\;e_{2}\phi_{3}\leq e_{2}^{2}+\frac{1}{4}\phi_{3}^{2}\\ e_{2}W_{2}(X_{2})=e_{2}[{℘}_{2}^{T}P_{2}(X_{2})+\psi_{2}(X_{2})]\leq 1/\left(4\mu_{2}^{2}\right)\beta_{2}e_{2}^{2}P_{2}^{T}P_{2}+\mu_{2}^{2}+\psi_{M}^{2}/4+e_{2}^{2}, \end{array}\right.$$
where $\mu_{2}$ represents the positive design parameter.

Integrating (54) into (53) produces: (55)$$\begin{array}{ll} {\dot{V}}_{2}\leq & -k_{1}e_{1}^{2}-\frac{1}{2}\sum_{i=3}^{4}\sum_{j=1}^{4}\xi_{ij}^{2}\left[x_{j}(t-\tau_{i})\right]-\left(\frac{1}{\lambda_{2}}-\frac{1}{2}\right)\phi_{2}^{2}+\left[1-16\tanh^{2}\left(\frac{e_{1}}{\sigma_{1}}\right)\right]T+\frac{\gamma_{1}}{d_{1}}{\tilde{\beta }}_{1}{\hat{\beta }}_{1}\\ & +e_{2}\left(e_{3}+u_{3}+\Delta \hat{E}-{\dot{u}}_{2c}\right)-ƛV_{T}+\phi_{3}{\dot{\phi }}_{3}+{\bar{M}}_{2}^{2}+\sum_{i=1}^{2}\mu_{i}^{2}++\frac{\psi_{M}^{2}}{2}+\frac{\phi_{3}^{2}}{4}\\ & +\frac{1}{4\mu_{2}^{2}}e_{2}^{2}\beta_{2}P_{2}^{T}P_{2}-\frac{1}{d_{2}}{\tilde{\beta }}_{2}{\dot{\hat{\beta }}}_{2}\;. \end{array}$$

Analogous to (38), choose the stunning control law $u_{3}$ and the adaptive law ${\dot{\hat{\beta }}}_{2}$ as: (56)$$\begin{array}{l} u_{3}=-\left(k_{2}e_{2}+\frac{1}{4\mu_{2}^{2}}{\hat{\beta }}_{2}e_{2}P_{2}^{T}P_{2}+\Delta \hat{E}\right)+{\dot{u}}_{2c}\\ {\dot{\hat{\beta }}}_{2}=\frac{d_{2}}{4\mu_{2}^{2}}e_{2}^{2}P_{2}^{T}P_{2}-\gamma_{2}{\hat{\beta }}_{2}, \end{array}$$
where the design parameters $k_{2}>0,\gamma_{2}>0$.

**Remark** **9.***Up to this step, the external interference* ΔE*can be thoroughly approximated by the DO (18) with the stunning controller (56), thus the stable operation of the controlled system is further enhanced.*

Pouring stunning control law $u_{3}$ and the adaptive law ${\dot{\hat{\beta }}}_{2}$ in (56) into (55) leads to: (57)$$\begin{array}{ll} {\dot{V}}_{2}\leq & -\sum_{i=1}^{2}k_{i}e_{i}^{2}-\frac{1}{2}\sum_{i=3}^{4}\sum_{j=1}^{4}\xi_{ij}^{2}\left[x_{j}(t-\tau_{i})\right]-\left(\frac{1}{\lambda_{2}}-\frac{1}{2}\right)\phi_{2}^{2}+\sum_{i=1}^{2}\frac{\gamma_{i}}{d_{i}}{\tilde{\beta }}_{i}{\hat{\beta }}_{i}+\left[1-16\tanh^{2}\left(\frac{e_{1}}{\sigma_{1}}\right)\right]T\\ & -ƛV_{T}+e_{2}e_{3}+\phi_{3}{\dot{\phi }}_{3}+{\bar{M}}_{2}^{2}+\sum_{i=1}^{2}\mu_{i}^{2}+\psi_{M}^{2}/2+\phi_{3}^{2}/4. \end{array}$$

Akin to (42), we can obtain: (58)$$\phi_{3}{\dot{\phi }}_{3}\leq -\left(\frac{1}{\lambda_{3}}-\frac{1}{4}\right)\phi_{3}^{2}+{\bar{M}}_{3}^{2},$$
where the function ${\bar{M}}_{3}>0$.

Substituting (58) into (57) produces: (59)$$\begin{array}{ll} {\dot{V}}_{2}\leq & -\sum_{i=1}^{2}k_{i}e_{i}^{2}-\frac{1}{2}\sum_{i=3}^{4}\sum_{j=1}^{4}\xi_{ij}^{2}\left[x_{j}(t-\tau_{i})\right]-\sum_{i=2}^{3}\left(\frac{1}{\lambda_{i}}-\frac{1}{2}\right)\phi_{i}^{2}+\sum_{i=1}^{2}\frac{\gamma_{i}}{d_{i}}{\tilde{\beta }}_{i}{\hat{\beta }}_{i}+\left[1-16\tanh^{2}\left(\frac{e_{1}}{\sigma_{1}}\right)\right]T\\ & -ƛV_{T}+e_{2}e_{3}+\sum_{i=1}^{2}\mu_{i}^{2}+\sum_{i=2}^{3}{\bar{M}}_{i}^{2}+\psi_{M}^{2}/2. \end{array}$$
**Step 3.** The design of the actual controller $u_{q}$ and the adaptive law ${\dot{\hat{\beta }}}_{3}$.

With the state error $e_{3}$ and estimation error ${\tilde{\beta }}_{3}$ considered, the third Lyapunov function $V_{3}$ is chosen as: (60)$$V_{3}=V_{2}+\frac{1}{2}e_{3}^{2}+\frac{1}{2d_{3}}{\tilde{\beta }}_{3}^{2},$$
where $d_{3}$ stands for the known positive constant.

Based on (24), derive $V_{3}$ in (60) as: (61)$${\dot{V}}_{3}={\dot{V}}_{2}+e_{3}{\dot{e}}_{3}-\frac{1}{d_{3}}{\tilde{\beta }}_{3}{\dot{\hat{\beta }}}_{3}.$$

Combining (23) and (59), (61)becomes: (62)$$\begin{array}{ll} {\dot{V}}_{3}\leq & -\sum_{i=1}^{2}k_{i}e_{i}^{2}-\frac{1}{2}\sum_{i=3}^{4}\sum_{j=1}^{4}\xi_{ij}^{2}\left[x_{j}(t-\tau_{i})\right]-\sum_{i=2}^{3}\left(\frac{1}{\lambda_{i}}-\frac{1}{2}\right)\phi_{i}^{2}+\sum_{i=1}^{2}\frac{\gamma_{i}}{d_{i}}{\tilde{\beta }}_{i}{\hat{\beta }}_{i}+\left[1-16\tanh^{2}\left(\frac{e_{1}}{\sigma_{1}}\right)\right]T+\frac{\psi_{M}^{2}}{2}\\ & -ƛV_{T}+e_{3}\left\{b_{1}x_{3}+b_{3}x_{2}+b_{2}x_{2}x_{4}+b_{4}u_{q}\;+\Delta l_{3}\left[x(t-\tau_{3})\right]+e_{2}-{\dot{u}}_{3c}\;\right\}+\sum_{i=1}^{2}\mu_{i}^{2}+\sum_{i=2}^{3}{\bar{M}}_{i}^{2}-\frac{1}{d_{3}}{\tilde{\beta }}_{3}{\dot{\hat{\beta }}}_{3}.\; \end{array}$$

Analogous to (30), the following inequality holds with n=4: (63)$$e_{3}\Delta l_{3}\left[x(t-\tau_{3})\right]\leq 2e_{3}^{2}+\frac{1}{2}\sum_{j=1}^{4}\xi_{3j}^{2}\left[x_{j}(t-\tau_{3})\right].$$

Pouring (63) into (62) leads to: (64)$$\begin{array}{ll} {\dot{V}}_{3}\leq & -\sum_{i=1}^{2}k_{i}e_{i}^{2}-\frac{1}{2}\sum_{j=1}^{4}\xi_{ij}^{2}\left[x_{j}(t-\tau_{i})\right]-\sum_{i=2}^{3}\left(\frac{1}{\lambda_{i}}-\frac{1}{2}\right)\phi_{i}^{2}+\sum_{i=1}^{2}\frac{\gamma_{i}}{d_{i}}{\tilde{\beta }}_{i}{\hat{\beta }}_{i}+\left[1-16\tanh^{2}\left(\frac{e_{1}}{\sigma_{1}}\right)\right]T+\frac{\psi_{M}^{2}}{2}\\ & -ƛV_{T}+e_{3}\left(b_{1}x_{3}+b_{3}x_{2}+b_{2}x_{2}x_{4}+b_{4}u_{q}\;+e_{2}+2e_{3}-{\dot{u}}_{3c}\;\right)+\sum_{i=1}^{2}\mu_{i}^{2}+\sum_{i=2}^{3}{\bar{M}}_{i}^{2}-\frac{1}{d_{3}}{\tilde{\beta }}_{3}{\dot{\hat{\beta }}}_{3}.\; \end{array}$$

Design the unknown function $W_{3}(X_{3})$ as: (65)$$W_{3}(X_{3})=b_{1}x_{3}+b_{2}x_{2}x_{4}+b_{3}x_{2}+e_{2}+3e_{3},$$
where $e_{2}=x_{2}-\alpha_{2c},e_{3}=x_{3}-\alpha_{3c}$. According to the composition of the unknown function $W_{3}(X_{3})$, it can be derived that $X_{3}$ consists of the unknown and uncertain states $x_{2},x_{3},x_{4},\alpha_{2c}$ and $\alpha_{3c}$. Then $X_{3}={\left[x_{2},x_{3},x_{4},u_{2c},u_{3c}\right]}^{T}$.

Consequently, rewrite (64) as: (66)$$\begin{array}{ll} {\dot{V}}_{3}\leq & -\sum_{i=1}^{2}k_{i}e_{i}^{2}-\frac{1}{2}\sum_{j=1}^{4}\xi_{ij}^{2}\left[x_{j}(t-\tau_{i})\right]-\sum_{i=2}^{3}\left(\frac{1}{\lambda_{i}}-\frac{1}{2}\right)\phi_{i}^{2}+\sum_{i=1}^{2}\frac{\gamma_{i}}{d_{i}}{\tilde{\beta }}_{i}{\hat{\beta }}_{i}+\left[1-16\tanh^{2}\left(\frac{e_{1}}{\sigma_{1}}\right)\right]T\\ & -ƛV_{T}+e_{3}[W_{3}(X_{3})\;+b_{4}u_{q}\;-e_{3}-{\dot{u}}_{3c}]+\sum_{i=1}^{2}\mu_{i}^{2}+\sum_{i=2}^{3}{\bar{M}}_{i}^{2}+\psi_{M}^{2}/2-1/d_{3}{\tilde{\beta }}_{3}{\dot{\hat{\beta }}}_{3}.\;\; \end{array}$$

Utilize an RBFNN to approximate $W_{3}(X_{3})$: (67)$$W_{3}(X_{3})={℘}_{3}^{T}P_{3}(X_{3})+\psi_{3}(X_{3}),\;\;\;\;\left|\psi_{3}(X_{3})\right|⩽\psi_{M}.$$

Resembling (36), it generates: (68)$$e_{3}W_{3}(X_{3})=e_{3}\left[{℘}_{3}^{T}P_{3}(X_{3})+\psi_{3}(X_{3})\right]\leq \frac{1}{4\mu_{3}^{2}}\beta_{3}e_{3}^{2}P_{3}^{T}P_{3}+\mu_{3}^{2}+\frac{\psi_{M}^{2}}{4}+e_{3}^{2},$$
where $\mu_{3}$ represents the positive design parameter.

Then, reperform (66) as: (69)$$\begin{array}{ll} {\dot{V}}_{3}\leq & -\sum_{i=1}^{2}k_{i}e_{i}^{2}-\frac{1}{2}\sum_{j=1}^{4}\xi_{ij}^{2}\left[x_{j}(t-\tau_{i})\right]-\sum_{i=2}^{3}\left(\frac{1}{\lambda_{i}}-\frac{1}{2}\right)\phi_{i}^{2}+\sum_{i=1}^{2}\frac{\gamma_{i}}{d_{i}}{\tilde{\beta }}_{i}{\hat{\beta }}_{i}+\left[1-16\tanh^{2}\left(\frac{e_{1}}{\sigma_{1}}\right)\right]T+\frac{3}{4}\psi_{M}^{2}\\ & -ƛV_{T}+e_{3}\left(b_{4}u_{q}-{\dot{u}}_{3c}\right)+\sum_{i=2}^{3}{\bar{M}}_{i}^{2}+\sum_{i=1}^{3}\mu_{i}^{2}+\frac{1}{4\mu_{3}^{2}}e_{3}^{2}\beta_{3}P_{3}^{T}P_{3}-\frac{1}{d_{3}}{\tilde{\beta }}_{3}{\dot{\hat{\beta }}}_{3}.\; \end{array}$$

Design the real controller $u_{q}$ and the adaptive law ${\dot{\hat{\beta }}}_{3}$ as: (70)$$\begin{array}{l} u_{q}=-\frac{1}{b_{4}}\left(k_{3}e_{3}+\frac{1}{4\mu_{3}^{2}}{\hat{\beta }}_{3}e_{3}P_{3}^{T}P_{3}-{\dot{u}}_{3c}\right)\\ {\dot{\hat{\beta }}}_{3}=\frac{d_{3}}{4\mu_{3}^{2}}e_{3}^{2}P_{3}^{T}P_{3}-\gamma_{3}{\hat{\beta }}_{3}, \end{array}$$
where the design parameters $k_{3}>0,\gamma_{3}>0$.

At this point, the actual controller for the q-axis has been designed.

Inserting the real controller $u_{q}$ and the adaptive law ${\dot{\hat{\beta }}}_{3}$ in (70) into (69) derives: (71)$$\begin{array}{ll} {\dot{V}}_{3}\leq & -\sum_{i=1}^{3}k_{i}e_{i}^{2}-\frac{1}{2}\sum_{j=1}^{4}\xi_{ij}^{2}\left[x_{j}(t-\tau_{i})\right]-\sum_{i=2}^{3}\left(\frac{1}{\lambda_{i}}-\frac{1}{2}\right)\phi_{i}^{2}+\sum_{i=1}^{3}\frac{\gamma_{i}}{d_{i}}{\tilde{\beta }}_{i}{\hat{\beta }}_{i}+\left[1-16\tanh^{2}\left(\frac{e_{1}}{\sigma_{1}}\right)\right]T\\ & -ƛV_{T}+\sum_{i=2}^{3}{\bar{M}}_{i}^{2}+\sum_{i=1}^{3}\mu_{i}^{2}+\frac{3}{4}\psi_{M}^{2}. \end{array}$$
**Step 4.** The design of the actual controller $u_{d}$ and the adaptive law ${\dot{\hat{\beta }}}_{4}$.

Consider the state error $e_{4}$ and the estimation error ${\tilde{\beta }}_{4}$, then the Lyapunov function $V_{4}$ is chosen as: (72)$$V_{4}=V_{3}+1/2e_{4}^{2}+1/\left(2d_{4}\right){\tilde{\beta }}_{4}^{2},$$
where $d_{4}$ represents the positive design parameter.

Combining (24), differentiate $V_{4}$ in (72) produces: (73)$${\dot{V}}_{4}={\dot{V}}_{3}+e_{4}{\dot{e}}_{4}-1/d_{4}{\tilde{\beta }}_{4}{\dot{\hat{\beta }}}_{4}.$$

Integrating (23) and (71) into (73) yields: (74)$$\begin{array}{ll} {\dot{V}}_{4}\leq & -\sum_{i=1}^{3}k_{i}e_{i}^{2}-\frac{1}{2}\sum_{j=1}^{4}\xi_{ij}^{2}\left[x_{j}(t-\tau_{i})\right]-\sum_{i=2}^{3}\left(\frac{1}{\lambda_{i}}-\frac{1}{2}\right)\phi_{i}^{2}+\sum_{i=1}^{3}\frac{\gamma_{i}}{d_{i}}{\tilde{\beta }}_{i}{\hat{\beta }}_{i}+\left[1-16\tanh^{2}\left(\frac{e_{1}}{\sigma_{1}}\right)\right]T\\ & -ƛV_{T}+e_{4}\left\{c_{1}x_{4}+c_{2}x_{2}x_{3}+\Delta l_{4}\left[x(t-\tau_{4})\right]+c_{3}u_{d}\right\}+3/4\psi_{M}^{2}+\sum_{i=2}^{3}{\bar{M}}_{i}^{2}+\sum_{i=1}^{3}\mu_{i}^{2}\\ & -\frac{1}{d_{4}}{\tilde{\beta }}_{4}{\dot{\hat{\beta }}}_{4}. \end{array}$$

Analogous to (30), the following inequality holds with n=4: (75)$$e_{4}\Delta l_{4}\left[x(t-\tau_{4})\right]\leq 2e_{4}^{2}+\frac{1}{2}\sum_{j=1}^{4}\xi_{4j}^{2}\left[x_{j}(t-\tau_{4})\right].$$

Design the unknown function $W_{4}(X_{4})$ as: (76)$$W_{4}(X_{4})=c_{1}x_{4}+c_{2}x_{2}x_{3}+3e_{4},$$
where $e_{4}=x_{4}$, the uncertain function $X_{4}$ consists of $x_{2},x_{3}$ and $x_{4}$. Then $X_{4}={\left[x_{2},x_{3},x_{4}\right]}^{T}$.

Then, (74) can be redrafted as: (77)$$\begin{array}{ll} {\dot{V}}_{4}\leq & -\sum_{i=1}^{3}k_{i}e_{i}^{2}-\sum_{i=2}^{3}\left(\frac{1}{\lambda_{i}}-\frac{1}{2}\right)\phi_{i}^{2}+\sum_{i=1}^{3}\frac{\gamma_{i}}{d_{i}}{\tilde{\beta }}_{i}{\hat{\beta }}_{i}+\left[1-16\tanh^{2}\left(\frac{e_{1}}{\sigma_{1}}\right)\right]T+\sum_{i=2}^{3}{\bar{M}}_{i}^{2}+\sum_{i=3}^{3}\mu_{i}^{2}\\ & -ƛV_{T}+e_{4}\left[W_{4}(X_{4})+c_{3}u_{d}-e_{4}\right]+3/4\psi_{M}^{2}-1/d_{4}{\tilde{\beta }}_{4}{\dot{\hat{\beta }}}_{4}. \end{array}$$

Similarly, the unknown function $W_{4}(X_{4})$ can be estimated by an RBFNN: (78)$$W_{4}(X_{4})={℘}_{4}^{T}P_{4}(X_{4})+\psi_{4}(X_{4}),\;\;\;\;\left|\psi_{4}(X_{4})\right|⩽\psi_{M}.$$

Pouring (78) into (77) leads to: (79)$$\begin{array}{ll} {\dot{V}}_{4}\leq & -\sum_{i=1}^{3}k_{i}e_{i}^{2}-\sum_{i=2}^{3}\left(\frac{1}{\lambda_{i}}-\frac{1}{2}\right)\phi_{i}^{2}-\frac{1}{d_{4}}{\tilde{\beta }}_{4}{\dot{\hat{\beta }}}_{4}+\sum_{i=2}^{3}{\bar{M}}_{i}^{2}+\sum_{i=1}^{3}\frac{\gamma_{i}}{d_{i}}{\tilde{\beta }}_{i}{\hat{\beta }}_{i}+\left[1-16\tanh^{2}\left(\frac{e_{1}}{\sigma_{1}}\right)\right]T\\ & -ƛV_{T}+e_{4}\left[-e_{4}+c_{3}u_{d}+{℘}_{4}^{T}P_{4}(X_{4})+\psi_{4}(X_{4})\right]+\sum_{i=1}^{3}\mu_{i}^{2}+3/4\psi_{M}^{2}. \end{array}$$

Analogous to (36), it derives: (80)$$e_{4}W_{4}(X_{4})=e_{4}\left[{℘}_{4}^{T}P_{4}(X_{4})+\psi_{4}(X_{4})\right]\leq \frac{1}{4\mu_{4}^{2}}e_{4}^{2}\beta_{4}P_{4}^{T}P_{4}+\mu_{4}^{2}+\frac{\psi_{M}^{2}}{4}+e_{4}^{2},$$
where $\mu_{4}$ stands for the positive design parameter.

Combining (80) with (79) generates: (81)$$\begin{array}{ll} {\dot{V}}_{4}\leq & -\sum_{i=1}^{3}k_{i}e_{i}^{2}-\sum_{i=2}^{3}\left(\frac{1}{\lambda_{i}}-\frac{1}{2}\right)\phi_{i}^{2}-\frac{1}{d_{4}}{\tilde{\beta }}_{4}{\dot{\hat{\beta }}}_{4}+\sum_{i=2}^{3}{\bar{M}}_{i}^{2}+\sum_{i=1}^{3}\frac{\gamma_{i}}{d_{i}}{\tilde{\beta }}_{i}{\hat{\beta }}_{i}+\left[1-16\tanh^{2}\left(\frac{e_{1}}{\sigma_{1}}\right)\right]T\\ & -ƛV_{T}+c_{3}e_{4}u_{d}+\sum_{i=1}^{4}\mu_{i}^{2}+\psi_{M}^{2}+\frac{1}{4\mu_{4}^{2}}e_{4}^{2}\beta_{4}P_{4}^{T}P_{4}. \end{array}$$

Devise the real controller $u_{d}$ and the adaptive law ${\dot{\hat{\beta }}}_{4}$ as: (82)$$\begin{array}{l} u_{d}=-\frac{1}{c_{3}}\left[k_{4}e_{4}+1/\left(4\mu_{4}^{2}\right){\hat{\beta }}_{4}e_{4}P_{4}^{T}P_{4}\right]\\ {\dot{\hat{\beta }}}_{4}=\frac{d_{4}}{4\mu_{4}^{2}}e_{4}^{2}P_{4}^{T}P_{4}-\gamma_{4}{\hat{\beta }}_{4}, \end{array}$$
where the design constants $k_{4}>0,\gamma_{4}>0$.

Fusing the real controller $u_{d}$ and the adaptive law ${\dot{\hat{\beta }}}_{4}$ in (82) into (81) produces: (83)$$\begin{array}{ll} \dot{V}\leq & -\sum_{i=1}^{4}k_{i}e_{i}^{2}-\sum_{i=2}^{3}\left(1/\lambda_{i}-1/2\right)\phi_{i}^{2}+\sum_{i=2}^{3}{\bar{M}}_{i}^{2}+\sum_{i=1}^{4}\gamma_{i}/d_{i}{\tilde{\beta }}_{i}{\hat{\beta }}_{i}\\ & -ƛV_{T}+\sum_{i=1}^{4}\mu_{i}^{2}+\psi_{M}^{2}+\left[1-16\tanh^{2}\left(e_{1}/\sigma_{1}\right)\right]T. \end{array}$$

At the very beginning of this subsection, a funnel variable (16) based on modified PPF is designed in this paper to guarantee that the tracking error narrows down to a prescribed funnel type scope. The modified PPF avoids the demand for the precise initial value of the output variable. Among these steps, RBFNNs are utilized to approximate the unknown hard-to-compute functions (33), (50), (65) and (76) based on their infinite approximation capability. The Lyapunov Krasovskii functional eliminates the time delays $\Delta l_{i}[x(t-\tau_{i})]$ in (3) with the aid of Lemma 1 and Remark 2. A DO (18) is introduced in step 2 to observe the uncertain matched and mismatched disturbance. 

Up to this step, the design process of the FDSC with time delays and output constraints is finished. The effectiveness of the controller and the boundedness of all the variables need to be discussed in Section 4 and proved in Section 5. 

## 4. Stability Analysis

For arbitrary predefined p>0, the compact sets are described as: (84)$$\;\left\{ \begin{array}{l} \Omega_{1}=\left\{\left(e_{1},\phi_{2},{\hat{\beta }}_{1},\xi_{11},\xi_{12},\ldots ,\xi_{44}\right):\;\;2\;\;e_{1}^{2}+2\phi_{2}^{2}+\frac{2}{d_{1}}{\tilde{\beta }}_{1}^{2}+4V_{T}\leq 4p\;\;\right\}\\ \Omega_{2}=\left\{\left(\begin{array}{l} e_{1},e_{2},\phi_{2},\phi_{3},{\hat{\beta }}_{1},{\hat{\beta }}_{2},\\ \xi_{11},\xi_{12},\ldots ,\xi_{44} \end{array}\right):\;\;2\sum_{i=1}^{2}e_{i}^{2}\;+2\sum_{i=2}^{3}\phi_{i}^{2}+\sum_{i=1}^{2}\frac{2}{d_{i}}{\tilde{\beta }}_{i}^{2}+4V_{T}\leq 4p\right\}\\ \Omega_{3}=\left\{\left(\begin{array}{l} e_{1},e_{2},e_{3},\phi_{2},\phi_{3},{\hat{\beta }}_{1},{\hat{\beta }}_{2},\\ {\hat{\beta }}_{3},\xi_{11},\xi_{12},\ldots ,\xi_{44} \end{array}\right):2\sum_{i=1}^{3}e_{i}^{2}+2\sum_{i=2}^{3}\phi_{i}^{2}+\sum_{i=1}^{3}\frac{2}{d_{i}}{\tilde{\beta }}_{i}^{2}+4V_{T}\leq 4p\right\}\\ \Omega_{4}=\left\{\left(\begin{array}{l} e_{1},\ldots ,e_{4},\phi_{2},\phi_{3},{\hat{\beta }}_{1},\ldots ,\\ {\hat{\beta }}_{4},\xi_{11},\xi_{12},\ldots ,\xi_{44} \end{array}\right):\;\;2\sum_{i=1}^{4}e_{i}^{2}+2\sum_{i=2}^{3}\phi_{i}^{2}+\sum_{i=1}^{4}\frac{2}{d_{i}}{\tilde{\beta }}_{i}^{2}+4V_{T}\leq 4p\right\}. \end{array}\right.$$

**Theorem** **1.***According to Assumption 1, the neural adaptive FDSC approach designed for the PMSM system (3) in this paper includes four controllers* $u_{2},u_{3},u_{q},u_{d}$*and four adaptive laws* ${\dot{\hat{\beta }}}_{i},i=1,\ldots ,4$*. For the initial conditions, if*$\Omega_{i},i=1,\ldots ,4$*,*$-f_{1}(0)<s_{1}(0)<f_{1}(0)$*and* $x_{d}\in (-d,d)$*are fulfilled, the core purposes of this paper will be realized.*

**Proof of Theorem** **1.**This part proves the boundedness and the binding ranges of the variables, demonstrating the effectiveness of the designed controller in this paper for the PMSM system. □

### 4.1. Verification of the Boundedness for All Variables

Considering the state errors $e_{i},i=1,\cdots ,4$, the filter errors $\phi_{i},i=1,2$, and the estimation errors ${\tilde{\beta }}_{i},i=1,\cdots ,4$, design the whole Lyapunov function V as: (85)$$V=V_{4}=1/2\sum_{i=1}^{4}e_{i}^{2}+1/2\sum_{i=2}^{3}\phi_{i}^{2}+\sum_{i=1}^{4}1/\left(2d_{i}\right){\tilde{\beta }}_{i}^{2}.$$

Then, it can be taken from (83) the derivative of V in (85) as: (86)$$\begin{array}{ll} \dot{V}\leq & -\sum_{i=1}^{4}k_{i}e_{i}^{2}-\sum_{i=2}^{3}\left(1/\lambda_{i}-1/2\right)\phi_{i}^{2}+\sum_{i=2}^{3}{\bar{M}}_{i}^{2}+\sum_{i=1}^{4}\gamma_{i}/d_{i}{\tilde{\beta }}_{i}{\hat{\beta }}_{i}\\ & -ƛV_{T}+\sum_{i=1}^{4}\mu_{i}^{2}+\psi_{M}^{2}+\left[1-16\tanh^{2}\left(e_{1}/\sigma_{1}\right)\right]T. \end{array}$$

Reorganize (86) as: (87)$$\dot{V}\leq -b_{0}V\;+\varphi_{0}+a_{1},$$
where $b_{0}=\min\{2k_{1},2k_{2},2k_{3},2k_{4},2\gamma_{1},2\gamma_{2},2\gamma_{3},2\gamma_{4},ƛ,(2/\lambda_{2}-1),\left(2/\lambda_{3}-1\right)\}$, $\varphi_{0}=\sum_{i=1}^{4}\mu_{i}^{2}+\sum_{i=2}^{3}{\bar{M}}_{i}^{2}$$+\psi_{M}^{2}$$,a_{1}=[1-16\tanh^{2}(e_{1}/\sigma_{1})]T$.

To ensure every term in (87) satisfying the stability conditions, it needs to choose proper design parameters that make up $b_{0}$ to guarantee $b_{0}>0$. With the aid of Lemma 3, the value of $a_{1}$ in (87) depends on two cases: (I) For $e_{1}\notin \Omega_{e_{1}}$, we can deduce that $a_{1}\leq 0$ due to T(x)≥0; (II) For $e_{1}\in \Lambda$, it can be inferred that $\left|e_{1}\right|\leq 0.2554\sigma_{1}$ and $\sigma_{1}>0$. Based on the above discussion, we can conclude that the boundedness of $e_{1}$ and $a_{1}$ are both assured. Consequently, a constant $\varphi_{1}>0$ can be chosen to fulfill $\left|\;\varphi_{0}+a_{1}\right|<\varphi_{1}$. Then redraft (87) as: (88)$$\dot{V}\leq -b_{0}V\;+\varphi_{1}.$$

Multiplying both sides by $\exp\left(b_{0}t\right)$, then (88) becomes $d(V(t)\exp\left(b_{0}t\right))/dt\leq \varphi_{1}\exp\left(b_{0}t\right)$. Integrating the inequality over [0,t], it leads to: (89)$$V(t)\leq \left[V(0)-\frac{\varphi_{1}}{b_{0}}\right]\exp\left(b_{0}t\right)+\frac{\varphi_{1}}{b_{0}}\leq V(0)+\frac{\varphi_{1}}{b_{0}}.$$

Especially, the following results can be derived from (89): (90)$$\underset{t\to \infty }{\lim}|e_{i}|\leq \sqrt{\frac{2\varphi_{1}}{b_{0}}},i=1,\ldots ,4.$$

It can be derived that $e_{i},\;\;\;i=1,\ldots ,4$ are bounded. Analogously, the boundedness of $\phi_{2},\phi_{3}$, and ${\tilde{\beta }}_{i},i=1,\ldots ,4$ are ensured with (85). From (24), it can be concluded that ${\hat{\beta }}_{i},i=1,\ldots ,4$ are bounded. We can know that $s_{1}$ is bounded from (16). Furthermore, it can be obtained that $x_{1}$ is bounded according to $s_{1}=x_{1}-x_{d}$ and the boundedness of $x_{d}$. Then, the virtual controller $u_{2}$ is bounded based on (38). Consequently, the boundedness of $x_{2}$ can be ensured from (20) and (21). Similarly, it can be derived that $u_{2},u_{d},u_{q},x_{i},i=2,3,4$ are also bounded. As a result, the designed controller demonstrates that all the signals of the closed-loop are uniformly bounded.

### 4.2. Deduction of the Binding Ranges for Variables

According to (85), one derives: (91)$$e_{i}^{2}\leq 2\left[V(0)-\frac{\varphi_{1}}{b_{0}}\right]e^{-b_{0}t}+\frac{2\varphi_{1}}{b_{0}},i=2,3,4.$$

Then we can get that $\left|e_{i}\right|\leq \sqrt{2[V(0)-\varphi_{1}/b_{0}]\exp(-b_{0}t)+2\varphi_{1}/b_{0}}$. $\left|e_{i}\right|\leq \sqrt{2\varphi_{1}/b_{0}}$ when $V(0)=\varphi_{1}/b_{0}$. If $V(0)\neq \varphi_{1}/b_{0}$, for arbitrary known $\sqrt{2[V(0)-\varphi_{1}/b_{0}]\exp(-b_{0}t)+2\varphi_{1}/b_{0}}>2\sqrt{\varphi_{1}/b_{0}}$, there exists T for each t>T, it produces $\sqrt{2[V(0)-\varphi_{1}/b_{0}]\exp(-b_{0}t)+2\varphi_{1}/b_{0}}$. When t→∞, the transformation error $\left|e_{i}\right|\leq \sqrt{2\varphi_{1}/b_{0}}$.

According to (16), $\eta_{1}\to \pm \infty$ when and only when $s_{1}\to \pm f_{1}$, it means that if $e_{1}\to \pm f_{1}$, $\Delta f_{i}$ will approach infinite. For the arbitrary starting value $\Delta f_{i}$, it satisfies $\left|s_{1}(0)\right|<f_{1}(0)$. For arbitrary t>0, it derives $\left|s_{1}(t)\right|<f_{1}(t)$. According to $s_{1}=x_{1}-x_{d}$, we can obtain the arbitrary initial condition $x_{d}(0)-f_{1}(0)<x_{1}(0)<x_{d}(0)+f_{1}(0)$ and other conditions $x_{1}\in \Pi_{x_{1}}:=\{x_{1}\in R:\;\;\;$$y_{d}(t)-f_{1}(t)<x_{1}(t)<y_{d}(t)+f_{1}(t)\}$ with t>0.

**Remark** **10.***The quality of the FDSC is judged by the value of* $e_{1}$*. Hence, it is of great importance to choose parameters based on the selection principle. The value of* $e_{1}$*depends on* $\varphi_{1}$*and* $b_{0}$. *When* $\varphi_{1}$*decreases and* $b_{0}$*increases,* $e_{1}$*will approach zero. Furthermore, we can increase* $\psi_{M},\Gamma_{1},\sum_{i=1}^{4}\mu_{i},\sum_{i=2}^{3}{\bar{M}}_{i}$*and decrease* $ƛ,\varepsilon_{2},1/\lambda_{2},1/\lambda_{3},k_{i},\gamma_{i},i=1,\cdots ,4$*in (87) to achieve excellent control performance of the system.*

## 5. Simulation and Comparison Results

### 5.1. Design of Controllers

To verify the effectiveness of the FDSC method, this subsection provides (3) with two simulation cases: (I) The case 1 contains time delays; (II) The case 2 ignores time delays ($\Delta f_{i}=0,\;i=1,2,3,4$). PID and NDSC approaches are used as comparison substrates in each case to more visually illustrate the superiority of the FDSC solution.

Select the inherent PMSM parameters as: $J=0.003798\mathrm{Kg}\cdot m^{2},B=0.001158\;N\cdot m/\left(\mathrm{rad}/s\right),$ $T_{L}=1.5,\;\varphi =0.1245\mathrm{Wb},\;L_{d}=0.00285H,\;n_{p}=3,\;L_{q}=0.00315H,R_{s}=0.68\Omega .$ The desired signal and the disturbance function are chosen as $x_{d}=0.02\sin(2t)+0.1$ and $\Delta E=40x_{2}\sin(2t)$, respectively. 

#### 5.1.1. FDSC

Based on Remark 10 and trial and error, the detailed FDSC’s design parameters as selected and shown in Table 2 to ensure the tracking error $e_{1}$ small enough: 

#### 5.1.2. NDSC

By virtual of [48], select the following variables of NDSC: (92)$$\left\{ \begin{array}{l} e_{1}=x_{1}-x_{d}\;\;\;\;\;e_{2}=x_{3}-\vartheta_{3c}\;\;\;\;\;\;e_{2}=x_{2}-\vartheta_{2c}\;\;\;\;\;\;e_{4}=x_{4}\\ {\dot{u}}_{2c}=1/\varepsilon_{2}\left(e_{2}-e_{2c}\right)\;\;\;\;\;\;\;\;\;\;\;\;\;\;\;\;{\dot{u}}_{3c}=1/\varepsilon_{3}\left(u_{3}-u_{3c}\right)\\ u_{2}=-k_{1}e_{1}+{\dot{x}}_{d}\;\;\;\;\;\;\;\;\;\;\;\;\;\;\;\;\;\;\;\;\;\;u_{3}=J/a_{1}\left(-k_{2}\ell_{2}+{\dot{u}}_{2c}-{\hat{\beta }}_{2}^{T}P_{2}\left(X_{2}\right)\right)\\ u_{q}=1/b_{4}\left(-k_{3}e_{3}+{\dot{\vartheta }}_{3c}-{\hat{\beta }}_{3}^{T}P_{3}\left(X_{3}\right)\right)\;\;\;\;\;\;u_{d}=J/c_{3}\left(-k_{4}e_{4}-{\hat{\beta }}_{4}^{T}P_{4}\left(X_{4}\right)\right)\\ {\dot{\hat{\beta }}}_{2}={\underline{\chi }}_{2}\left[P_{2}\left(X_{2}\right)e_{2}-\gamma_{2}{\hat{\beta }}_{2}\right]\;\;\;\;\;\;{\dot{\hat{\beta }}}_{3}={\underline{\chi }}_{3}\left[P_{3}\left(X_{3}\right)e_{3}-\gamma_{3}{\hat{\beta }}_{3}\right]\\ {\dot{\hat{\beta }}}_{4}={\underline{\chi }}_{4}\left[P_{4}\left(X_{4}\right)e_{4}-\gamma_{4}{\hat{\beta }}_{4}\right], \end{array}\right.$$
based on Remark 10 and trial and error, the design parameters are chosen to guarantee the tracking error small enough: $k_{1}=30,k_{2}=k_{3}=k_{4}=80,{\underline{\chi }}_{2}={\underline{\chi }}_{3}={\underline{\chi }}_{4}=10,\gamma_{2}=\;\gamma_{3}=\gamma_{4}=0.09,\lambda_{2}=$$\lambda_{3}=0.01$.

#### 5.1.3. PID

According to [12], the following PID controller was chosen with $u_{d}=0$: (93)$$u_{q}=k_{p}s_{1}+k_{i}\int_{0}^{t}s_{1}d\tau +k_{d}d\left(s_{1}\right)/dt,$$
based on Remark 10 and trial and error, the design parameters are chosen to guarantee the tracking error small enough: $k_{p}=20,k_{i}=0.05,k_{d}=1.5$.

Meanwhile, Table 3 lists three quantitative indicators to compare FDSC scheme with the NDSC and PID methods.
(a)Integration over the absolute value of the error (IAE):
(94)$$J_{IAE}=\int_{0}^{t}\left|\ell (\tau )\right|d\tau .$$
(b)Integration over time and the absolute value of the error (ITAE):
(95)$$J_{ITAE}=\int_{0}^{t}\tau \left|\ell (\tau )\right|d\tau .$$
(c)Integration over squared error (ISE):


(96)
$$J_{ISE}=\int_{0}^{t}\ell^{2}(\tau )d\tau .$$


Upon the above discussions, consider the two cases as:

**Case 1:** Select the time-delay terms as: (97)$$\left\{ \begin{array}{l} \Delta f_{1}(x(t-\tau_{1}))=10x_{1}^{4}(t-\tau_{1})x_{2}(t-\tau_{1})x_{3}(t-\tau_{1})x_{4}(t-\tau_{1})\;\;\;\\ \Delta f_{2}(x(t-\tau_{2}))=12x_{1}^{4}(t-\tau_{2})x_{2}^{4}(t-\tau_{2})x_{3}(t-\tau_{2})x_{4}(t-\tau_{2})\\ \Delta f_{3}(x(t-\tau_{3}))=14x_{1}^{4}(t-\tau_{3})x_{2}^{4}(t-\tau_{3})x_{3}^{4}(t-\tau_{3})x_{4}(t-\tau_{3})\\ \Delta f_{4}(x(t-\tau_{4}))=16x_{1}^{4}(t-\tau_{4})x_{2}^{4}(t-\tau_{4})x_{3}^{4}(t-\tau_{4})x_{4}^{4}(t-\tau_{4}). \end{array}\right.$$

Design the time-varying funnel type boundaries as $f_{1}(t)=e^{-2t}+0.1t/(2t+2),$ and select the original values of the state variables as $x_{i}(0)=0.01,\;\;\;\;\;i\;=1,\ldots ,4.$ Each RBFNN consists of 11 nodes with the center positioned in the interval [−11,11], and the width is 10. 

**Case 2****:** The time-delay terms are $\Delta f_{i}=0,\;i=1,2,3,4.$ In other words, the time-delays are not considered.

### 5.2. Simulation Comparison Results

The simulation comparison results are shown in Figure 3, Figure 4, Figure 5, Figure 6 and Figure 7, every figure contains two sub-figures: Case 1 and Case 2. The simulation results will prove the superiority of the FDSC scheme, and justify whether the Lyapunov-Krasovskii functional is effective.

It can be seen from Figure 3 that the FDSC method ensures the best convergence of output signal $x_{1}$ when compared with PID and NDSC methods no matter in Case 1 or Case 2. And the output signal $x_{1}$ based on the FDSC method is limited with the funnel prescribed boundaries when the output signal $x_{1}$ based on the NDSC method is out of the boundary. The curve of output signal $x_{1}$ in case 1 is almost the same as it in case 2, which means that Lyapunov-Krasovskii functional can effectively address the time delays.

It is demonstrated from Figure 4 that the FDSC method has the smallest tracking error $s_{1}$ among three control schemes no matter in Case 1 or Case 2. And the tracking error $s_{1}$ based on the FDSC method is limited by the funnel prescribed boundaries when compared with the NDSC method. The curves of tracking error $s_{1}$ are almost the same in the two cases, showing the effectiveness of the Lyapunov-Krasovskii functional.

It is shown from Figure 5 that the curve of the state variable $x_{2}$ in Case 1 is almost the same as it in Case 2. It means that the Lyapunov-Krasovskii functional can effectively address the time delays. The controller designed in is paper guarantees that the state variable $x_{2}$ in the closed-loop is controllable. The same conclusion can be drawn in Figure 6.

It can be seen from Figure 6 that the curves of the state variables $i_{d}$ and $i_{q}$ in case 1 are almost the same as them in case 2. It means that the Lyapunov-Krasovskii functionals can effectively address the time delays. The controller designed in is paper guarantees the stable operation of the PMSM systems with time delays. And the d− axis current $i_{d}$ and the q− axis current $i_{q}$ in the closed-loop are controllable. The same conclusion can be drawn in the Figure 7.

Within the funnel-type boundaries, Figure 3 gives the curves of output signal $x_{1}$ and ideal signal $x_{d}$ under three schemes (FDSC, PID, and NDSC). Figure 4 exhibits the curves of tracking error $s_{1}$ for different scenarios. It can be concluded from Figure 3 and Figure 4 that FDSC has a better convergency rate than NDSC, and the tracking error $s_{1}$ of FDSC method is minimal compared to the other two options without breaking through funnel-type boundaries even considering the time delays. Figure 5 displays the curves of state $x_{2}$ in the FDSC approach. Figure 6 illustrates the trajectories of states $i_{d}$ and $i_{q}$, and Figure 7 shows the curves of controllers $u_{d}$ and $u_{q}$. According to Figure 3, Figure 4, Figure 5, Figure 6 and Figure 7, the states of the FDSC have extremely rapid response capability compared to NDSC and PID with time delays.

## 6. Conclusions

The FDSC with disturbance-observer has been investigated and suggested to resolve the position tracking control challenge for the PMSM with time delays here. At the beginning of this paper, an assumption and several lemmas are employed to uphold the control thesis and the design procedure of the funnel controller. With the aid of them, the inequalities in Section 3 can be simplified. Secondly, this paper introduces promising tools to eliminate the nonlinear uncertainties in PMSMs. For instance, this paper introduces RBFNNs to estimate the unknown functions because they can approximate the unknown hard-to-calculated functions that are in a closed set at any precision. For the unknown mismatched and matched external disturbance, this paper utilizes a finite-time second-order DO. The finite-time DO can approximate the unknown interference with a finite time speed. To remove the limitations of the output variable, a funnel variable is designed in this paper without the demand for precise initial values of the state variable. This paper uses the first-order filters to filtrate the multiple derivatives of the virtual controllers so that the “explosion of complexity” can be addressed. The Lyapunov-Krasovskii functionals are devised in the controller design procedure of this paper to deal with time delays and guarantee the stable operation of the PMSMs. Thirdly, with the aid of RBFNNs, the finite-time DO, first-order filters, and the Lyapunov-Krasovskii functionals, the controller design procedure can be finished based on the backstepping method. Despite that the devised FDSC method can provide outstanding tracking performance shown in Figure 3, Figure 4, Figure 5, Figure 6 and Figure 7 for the output-constrained PMSM, the finite-time response rates need to be further improved by combining it with finite-time stability theory. In further research later, we will further study how to improve the finite time convergence of tracking error based on the FDSC scheme. For PMSMs are widely applied in many significant fields, the neural adaptive FDSC scheme can be extended to vehicles, electric elevators, and machine tools.

## Figures and Tables

**Figure 1 entropy-24-01028-f001:**
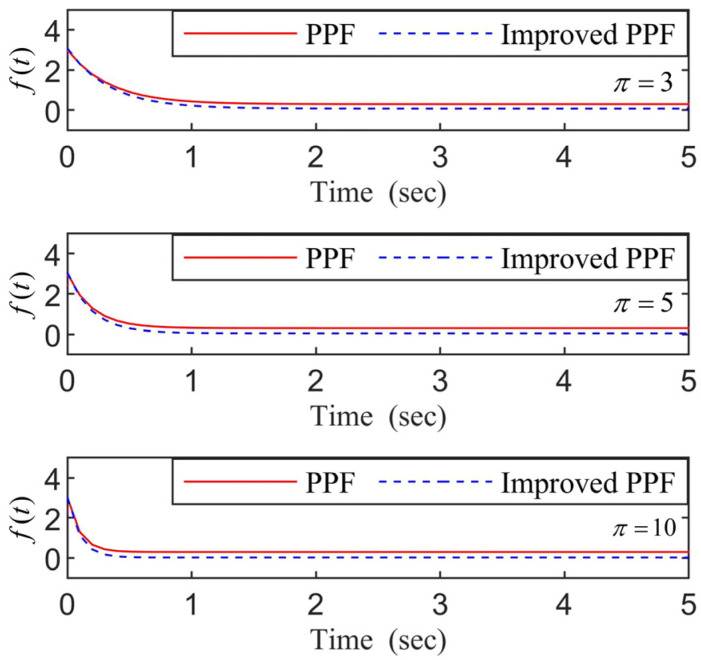
The curves of PPF and improved PPF (π=3,5,10).

**Figure 2 entropy-24-01028-f002:**
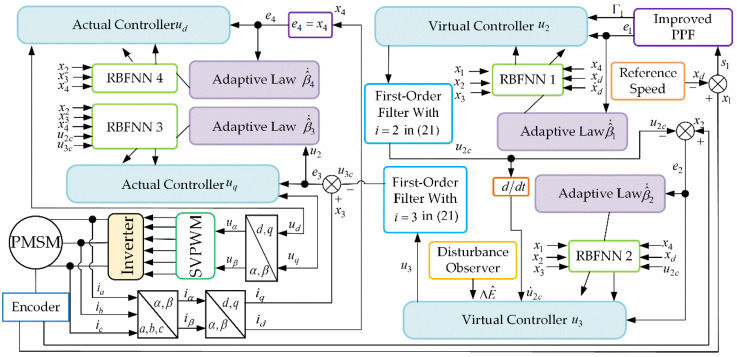
Overview of the control architecture for the PMSM system.

**Figure 3 entropy-24-01028-f003:**
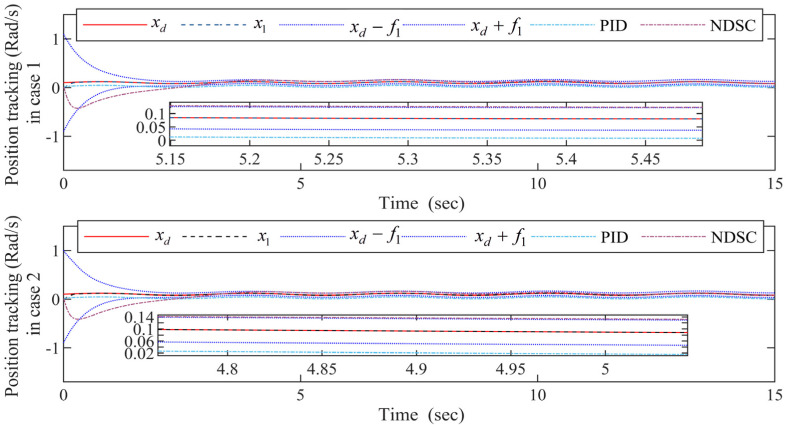
The output signal $x_{1}$ and the ideal signal $x_{d}$ curves for the FDSC, PID, and NDSC.

**Figure 4 entropy-24-01028-f004:**
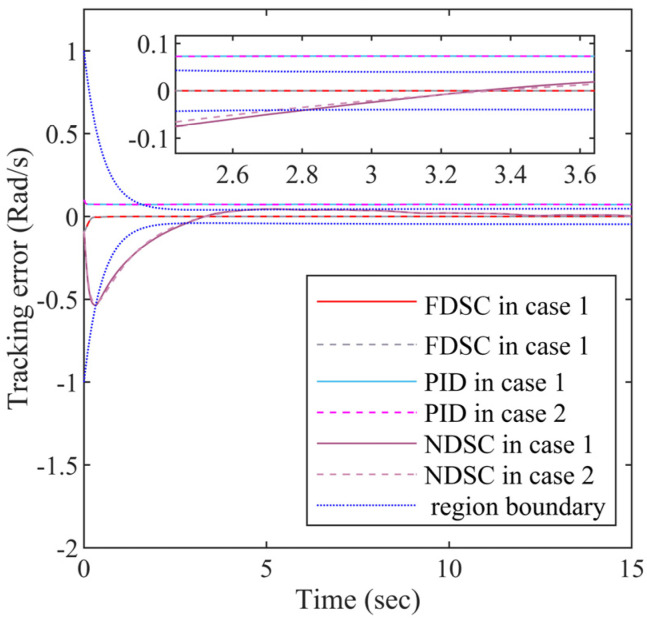
The tracking error responses $s_{1}$ for the FDSC, PID, and NDSC.

**Figure 5 entropy-24-01028-f005:**
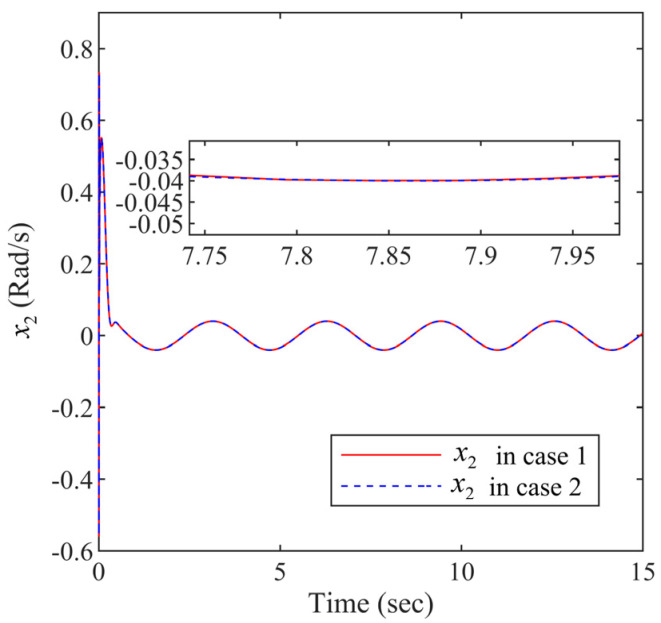
The state variable $x_{2}$ responses.

**Figure 6 entropy-24-01028-f006:**
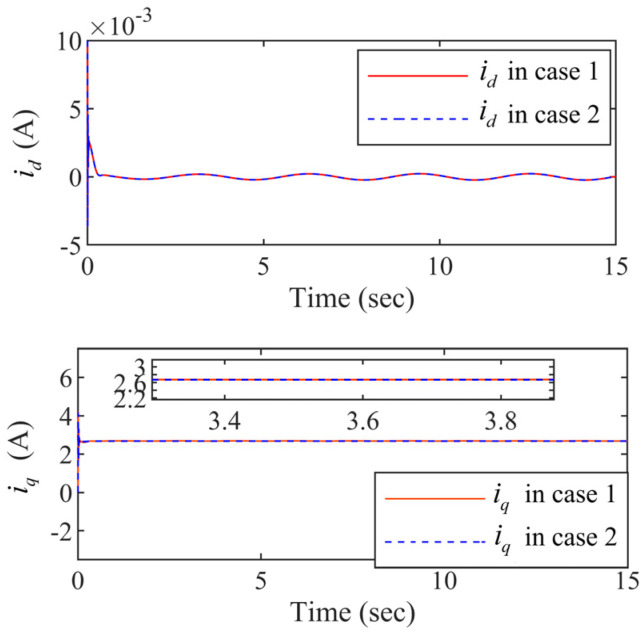
The state variables $i_{d}$ and $i_{q}$ responses.

**Figure 7 entropy-24-01028-f007:**
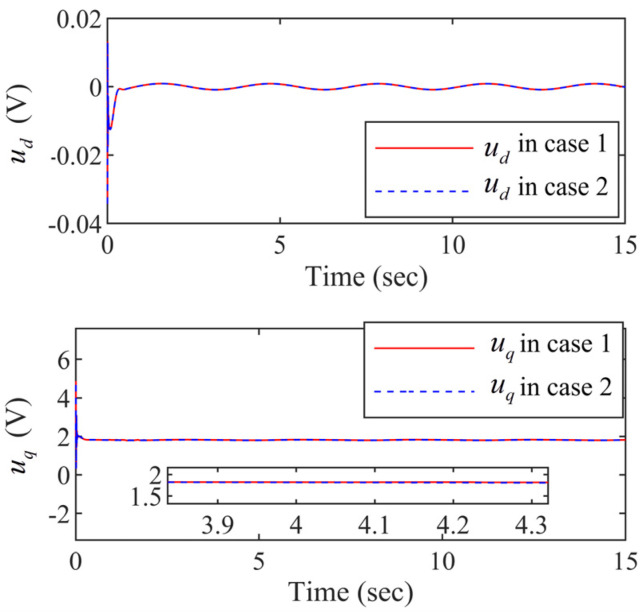
The controllers $u_{d}$ and $u_{q}$ trajectories.

**Table 1 entropy-24-01028-t001:** Parameters of PMSM.

Parameters	Descriptions	Units
θ	Rotor angular	rad
ω	Rotor angular velocity, $\dot{\theta }$	rad/s
$\dot{\omega }$	First-order derivative of rotor angular velocity	${\mathrm{rad}/s}^{2}$
$i_{d},i_{q}$	Currents of d−q axis	A
$u_{d},u_{q}$	Voltages of d−q axis	V
$L_{d},L_{q}$	Stator inductances of d−q axis	H
J	Inertia rotor moment	$\mathrm{kg}\cdot m^{2}$
B	Friction coefficient	N·m/(rad/s)
φ	Inertia magnet flux linkage	Wb
$R_{s}$	Armature resistance	Ω
$n_{p}$	Pole pairs	
$T_{L}$	Load torque	N·m

**Table 2 entropy-24-01028-t002:** FDSC’s design parameters.

Parameters	Values	Parameters	Values	Parameters	Values
$u_{2c}(0)$	0	${\hat{\beta }}_{1}\;(0)$	−0.05	$d_{1}$	0.65
$u_{3c}(0)$	0.5	${\hat{\beta }}_{2}\;(0)$	0	$d_{2}$	0.95
$\varepsilon_{2}$	0.1	${\hat{\beta }}_{3}\;(0)$	−0.5	$d_{3}$	0.75
$\varepsilon_{3}$	0.01	${\hat{\beta }}_{4}\;(0)$	0	$d_{4}$	35
$k_{1}$	10	$\gamma_{1}$	60	$\mu_{1}$	0.06
$k_{2}$	20	$\gamma_{2}$	4	$\mu_{2}$	0.3
$k_{3}$	20	$\gamma_{3}$	60	$\mu_{3}$	0.1
$k_{4}$	1200	$\gamma_{4}$	0.4	$\mu_{4}$	0.01

**Table 3 entropy-24-01028-t003:** Comparative numerical results of performance indicators.

Indicators		FDSC	PID	NDSC
Case 1	ISE	0.000661	0.079390	0.244500
	ITAE	0.005941	8.171000	2.801000
	IAE	0.012980	1.091000	0.978600
Case 2	ISE	0.000661	0.079010	0.247000
	ITAE	0.005894	8.151000	2.764000
	IAE	0.012980	1.089000	0.970700

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
