# Peer review of "Neural Adaptive Funnel Dynamic Surface Control with Disturbance-Observer for the PMSM with Time Delays"

_entropy, 2022, doi:10.3390/e24081028_

Round 1

Reviewer 1 Report

Please see the attached PDF file for the detailed review.

Author Response

Dear Reviewer,

We have seriously revised the manuscript (Manuscript ID: entropy-1806262), originally entitled Neural Adaptive Funnel Dynamic Surface Control with Disturbance-Observer for the PMSM with Time Delays”. We greatly appreciate your quick feedback and constructive comments. The point-by-point responses to the nice reviewer are carefully addressed and explained below.

We sincerely hope that our reply and revision would completely satisfy the reviewers.

We look forward to hearing from you.

With the best regards,

All Authors.

Reviewer 2 Report

In this paper, the authors presented a neural adaptive funnel dynamic surface control methods for permanent magnet synchronous motors with time delays. the paper is slightly unbalanced with very little discussions on the results, lead to some doubt on the method that authors proposed. the suggested changes are:

1. the contribution from the authors, i.e. what is actually achieved is not clear.

2. in equation1 dot theta, dot omega is not defined.

3. there are quite a few parameters described in the paper, suggest to create a nomenclature section for it.

4. figure 2 should brought forward before the description of the steps

5. in section 5.1 and perhaps other sections as well, please justify the parameters selected, same as the parameters in Table 2.

6. there are lack of detailed analysis of the simulation results and section 5.2 should be expanded.

7. conclusion is short and should rewrote.

Author Response

(The authors gave the same response as above.)

Round 2

Reviewer 1 Report

Hyphenation in the paper's title doesn't look good (Dis-turbance).

The unit symbols should be in normal (roman) font.  Currently they are still in italic (cursive) font - see Table 1.

"Error! Reference source not found" in page 10.

Reviewer 2 Report

The authors have addressed the reviewers comments and no further changes are required.